# Trust in government moderates the association between fear of COVID-19 as well as empathic concern and preventive behaviour

With the COVID-19 pandemic, behavioural scientists aimed to illuminate reasons why people comply with (or not) large-scale cooperative activities. Here we investigated the motives that underlie support for COVID-19 preventive behaviours in a sample of 12,758 individuals from 34 countries. We hypothesized that the associations of empathic prosocial concern and fear of disease with support towards preventive COVID-19 behaviours would be moderated by trust in the government. Results suggest that the association between fear of disease and support for COVID-19 preventive behaviours was strongest when trust in the government was weak (both at individual- and country-level). Conversely, the association with empathic prosocial concern was strongest when trust in the government was high, but this moderation was only found at individual-level scores of governmental trust. We discuss how motivations may be shaped by socio-cultural context, and outline how findings may contribute to a better understanding of collective action during global crises.

---

The official proclamation of the global coronavirus (COVID-19) pandemic in March 2020[1] was followed by the institution of a variety of strategies to limit the spread of the virus among governments all around the world. For instance, in many places, lockdowns were initiated, masks were mandated, social distancing enforced, while vaccination and isolation requirements were imposed[2]. However, many citizens rallied against these public health mandates and recommendations, motivated by pseudoscientific claims and conspiratorial beliefs. Public mistrust in governmental measures and policies grew as a consequence, while many refused to get a vaccine, or practice safe and preventive behaviors to contain the spread of the virus (including physical distancing, hand washing, or face covering)[3,4].

The scientific quest to illuminate reasons why citizens across the world support or undermine safe and protective behaviors, came as a natural response to the global pandemic. Current evidence suggests that individuals may support or practice behaviors that prevent the spread of COVID-19 either for their self-interest (i.e., fear of becoming infected)[5–12], or for other-oriented prosocial reasons (i.e., fear of infecting others)[13–16]. Yet, a deeper understanding of how these motives may interact with their socio-cultural conditions is lacking.

Such understanding is relevant to identify effective ways to mobilize support for COVID-19 preventive practices. In particular, research would benefit from gaining deeper insight on how self- versus other-oriented concerns affect attitudes towards preventive behaviors of people with different levels of governmental trust (i.e., individuals with high versus low trust in the government) and people living in different socio-cultural contexts (where trust in the government is typically high versus low). The present study addresses this call and tests how trust in the government (both on individual- and on country-level) may moderate the relationship between self-oriented fear and other-oriented empathic concerns regarding COVID-19 disease, with supportive attitudes toward the practice of COVID-19 containment behaviors.

Mobilizing people to engage in large-scale collective activities is a complex, multifaceted, and intricate problem. A core challenge may be the often uncertain or insufficient incentives for truly global cooperative activities. The incentives for large-scale cooperation do not become evident immediately and the level of unpredictability is typically high. Namely, in large-scale cooperation it is often unclear whether one's contribution will be tracked, rewarded, reciprocated, or will produce any large-scale benefit[17,18]. All of these are conditions that can aggravate cooperation, especially if it only serves altruistic goals.

However, cooperation does not only rely on altruistic reasons, but can also be driven by selfish motives[17,18], and this is corroborated by evidence from the latest pandemic. Namely, compliance with COVID-19 mitigating behaviors is associated with both other-oriented concerns such as empathy and altruism[5,7,10,19] and self-oriented concerns such as fear of infection[9,14,20]. However, it is less clear how contextual conditions will shape these motives, and under which conditions they will be more (or less) strongly associated with COVID-19 compliance. For instance, research with samples from Sweden, Germany, United States of America (USA) and Norway suggests that other-oriented concerns were more influential than fear of COVID-19 in promoting adherence to containment measures[6,13,21,22]. On the other hand, a study by Zirenko and colleagues found that caring for oneself played a more important role than caring for others when predicting individuals' decisions to physically distance in Russia, Azerbaijan, and China[23]. Hence, individuals in different contexts may all (more or less) comply with COVID-19 containment measures, yet their reasons to do so

may be different depending on relevant (perceived) contextual conditions, with trust in the government being one of the most important factors.

Cooperation requires trust, especially when it involves unrelated others[24–26] and when it revolves around large-scale collaborations. People are particularly prone to coalesce in larger groups if they are motivated by trust toward key authority figures who regulate social norms[27] like government officials, state representatives, or renowned politicians[28–30]. It is thus likely that governmental trust is crucial in motivating joint cooperative action to contain the COVID-19 pandemic. Yet, a notion that higher trust in the government is associated with higher (likelihood for) engagement in COVID-19 containment behaviors may not hold true unconditionally, despite the emergence of some supportive evidence[31–34]. There is also evidence to suggest that the relationship between trust in the government and cooperative behaviors during the COVID-19 pandemic is less clear and seems more complex[35,36]. For instance, research by Clark and colleagues[37] with data collected from a large international sample found that trust in the government was not related to how much people reported to adhere with governmental recommendations, while it showed weak associations with taking private health precautions. Hence, in the present study, we conceptualized trust in the government as a boundary condition that moderates the association between support for COVID-19 containment behaviors with other-oriented (i.e., empathic prosocial concern) and self-oriented (i.e., fear of COVID-19) motives.

As outlined above, mobilizing people for large-scale and long-term cooperative acts is challenging; mainly due to the high level of uncertainty it involves. Some individuals (especially those who feel less vulnerable to the virus) might even experience a social dilemma whereby cooperating would require them to sacrifice their concrete, short-term, and selfish interests (e.g., by avoiding social events, by face masking, etc.) over the more abstract, less traceable, and long-term goal of protecting other people from infection. We argue that such a dilemma should become especially salient, and thus hinder empathy-driven cooperation, under highly unpredictable conditions where one is doubtful about whether or not individual sacrifices will promote any long-term public benefit. The way in which people perceive their governments, namely whether they view them as competent, protective, and caring – simply, whether they trust in their government or not – may represent one of the factors that guide people's perceptions of (un)predictability. Namely, when people anticipate their governments to be supportive and to take the needed measures for protecting their citizens from any threat, individuals would become more confident that their selfless and empathy-driven activities will truly result in positive public health outcomes. On the contrary, when there is little trust in the government, unpredictability will increase, and individuals will doubt whether their individual compliance with public health measures will produce any public, other-oriented benefit. Even if respective measures are implemented by the government, those with little trust would be propelled to suspect that they may be implemented in a superficial and non-transparent manner.

However, under such volatile conditions cooperation may still occur, yet for different sets of reasons. The compliance with safe and preventive COVID-19 behaviors is not only an act that may be performed for the sake of protecting others from infection, but also has direct (beneficial) implications on personal health and may thus be considered as a self-protective act. In circumstances where trust in the government is low, people would experience higher uncertainty, feel more susceptible to the virus, and their cooperation would be more strongly driven by selfish concerns such as fear of COVID-19. Therefore, we propose that the relationship between empathic concern and support for COVID-19

containment behaviors will be stronger when the government is (generally) perceived as more (compared to less) trustworthy, and that conversely the association between fear of disease and support for COVID-19 containment behaviors will be stronger when the government is (generally) perceived as less (compared to more) trustworthy.

The current study tests its premises with a large, cross-national sample of 12,758 individuals recruited across 34 countries which adds to its robustness, as cooperation against the COVID-19 pandemic is not only determined by individual factors, but also shaped by the different socio-cultural conditions that exist across various societies. Such factors range from more general cultural and developmental differences such as differences in the content and strictness of social norms, the level of education and affluence; to more specific COVID-19 associated differences such as differences in the health care system, the stringency of government policies, and the severity level of the COVID-19 pandemic. To embrace such diversity, the 34 nations involved in the present research consist of both highly developed and affluent countries, as well as less-developed and relatively poor countries[38], and represent both countries that were severely affected by the COVID-19 pandemic at time of data collection, as well as countries that were hardly affected by the pandemic (Table 1). Hence, nested within such different cultural and sociopolitical contexts, the present study proposes and tests the following pre-registered hypotheses (https://osf.io/k2wjr[39]).

Empathic prosocial concern is more strongly associated with support for COVID-19 containment behaviors (e.g., physical distancing, face masking, enhanced hygiene practices) among individuals with higher levels of trust in the government compared to individuals with lower levels of trust (Hypothesis 1a). Conversely, fear of COVID-19 is more strongly associated with the support for COVID-19 containment behaviors among individuals with low levels of trust in the government compared to individuals with higher levels of trust (Hypothesis 2a).

Congruent with testing these two hypotheses on an individual level, we further propose that country-level trust scores will moderate the association between support for COVID-19 containment behaviors with empathy and fear motivations in such a way that individuals' empathic concern is more strongly related to supporting COVID-19 containment behaviors in contexts where trust in the government is generally high compared to contexts where trust is generally low (Hypothesis 1b); and that individuals' fear of COVID-19 is more strongly related to supporting COVID-19 containment behaviors in contexts where trust in the government is generally low compared to contexts where trust is generally high (Hypothesis 2b).

## Methods
The current study protocol has been reviewed and approved by the Institutional Review Board of Bahcesehir University (IRB protocol number: E-8755). When not declared as exempt, approvals have additionally been obtained from the local institutional review boards of all other involved countries. All ethical guidelines were followed when conducting the present research. Written informed consent to take part in the study was obtained from each respondent prior to completing the research, and only respondents who agreed to the study's conditions were allowed to proceed with the research.

The present research was first preregistered on September, 9th, 2021 under https://osf.io/k2wjr[39]; and has been updated on March 29th, 2023, for the following reasons and aspects, respectively. First, reviewers requested to avoid any directional language in the hypotheses and to align the variable names more closely with the measurement procedure and the labeling in the

manuscript. For this reason, the wording of the hypotheses has been revised. The original hypotheses were preregistered as: Governmental trust will moderate the effect of other-centered motives on performing virus containment behaviors (socially responsive COVID-19 behaviors and vaccination intentions) in such a way that the effect will be stronger for respondents holding high compared to low levels of trust (Hypothesis 1); and governmental trust will moderate the effect self-centered motives on performing virus containment behaviors, in such a way that the effect will be stronger for respondents holding low compared to high levels of trust (Hypothesis 2).

Second, the reviewers and the editor asked to preregister a more concrete data analysis plan. Specifically, they asked for adding relevant individual-level (e.g., gender and age) and country-level control variables (e.g., number of COVID-19 cases) when conducting the regression analysis. Hence, the analyses of the present research were conducted by controlling for relevant covariate effects. Further, the reviewers requested to conduct the same analyses with generalized trust (both at individual and country-level) to examine whether possible moderator effects are unique to trust in the government or rather related to a more general notion of trust. The results concerning these analyses are presented under exploratory analysis. And finally, the reviewers asked to check the robustness of the results against using a fixed effects regression model with cluster-robust standard errors. The results concerning this alternative model are presented under Supplementary Note 1 and Supplementary Table 1. The preregistered analysis plan has been updated to meet these requests.

**Procedure**. The present study was spearheaded by the Research Initiatives Working Group (RIWG) of the American Psychological Association (APA) Interdivisional Task Force on the Pandemic, committed to the advancement of a knowledge base through a repository and dissemination of materials and resources[40]. The data collection was conducted within the framework of a large-scale collaborative project spanning across different nations around the globe, which is named as International and Multidimensional Perspectives on the Impact of COVID-19 across Generations (IMPACT-C19). The project focused specifically on the impact, perceptions, and experiences of COVID-19 among young people and established adults in an international perspective[41]. Participating researchers were invited to collect self-report online data that were created with survey tools such as Google Forms or Qualtrics using the convenience sampling method.

**Participants**. In accordance with our sampling strategy, the minimal number of participants was approximately 150 adult respondents per country. Namely, we conducted an a-priori power analysis using G*Power 3.1.9[42], assuming a medium effect size of $f^2 = 0.10$, targeting a power of 0.95 and an alpha level of 0.05, which suggested a sample size of 158 per collection site (country).

The raw dataset without any data exclusions by January 20th, 2022, comprised data of 27,787 responses. We first deleted responses from all respondents that were aged below 18 or did not indicate any age ($N = 4951$); and then list-wise excluded all data with any missing values on the study variables ($N = 8161$). Finally, data from 22 countries with less than 150 complete responses were removed ($N = 1917$) resulting in the final dataset based on which the analyses were performed. These countries were Czech Republic, Slovenia, Taiwan, USA, Costa Rica, Niger, Zambia, Zimbabwe, Afghanistan, Dominican Republic, Uganda, Mozambique, Argentina, Kazakhstan, Kosovo, Albania, North Macedonia, Armenia, Guatemala, Bosnia Herzegovina, Qatar,

**Table 1 Cultural, socio-political and COVID-19 associated differences between the countries of the present study.**

| | Human Development Index (HDI) (2021) | | | | | Hospital beds per 1000, 2017–2020 | Month of data collection start in 2021 | Government Stringency level | Daily new confirmed COVID-19 cases | Daily new confirmed COVID-19 deaths | Cultural tightness |
|---|---|---|---|---|---|---|---|---|---|---|---|
| | HDI Total Score | Life expectancy at birth | Expected years of schooling | Mean years of schooling | Gross National Income (GNI) per capita in USD | | | | | | |
| Australia | 0.944 | 83.4 | 22.0 | 12.7 | 48,085 | 3.84 | April | 46.76 | 0.52 | 0 | −0.05 |
| Bangladesh | 0.632 | 72.6 | 11.6 | 6.2 | 4976 | 0.79 | April | 83.33 | 31.32 | 0.27 | – |
| Brazil | 0.765 | 75.9 | 15.4 | 8.0 | 14,263 | 2.09 | March | 67.13 | 262.23 | 5.72 | −0.38 |
| Bulgaria | 0.816 | 75.1 | 14.4 | 11.4 | 23,325 | 7.45 | March | 53.70 | 228.58 | 7.77 | – |
| Colombia | 0.767 | 77.3 | 14.4 | 8.5 | 14,257 | 1.71 | Sept. | 46.30 | 38.82 | 1.51 | −0.58 |
| Croatia | 0.851 | 78.5 | 15.2 | 11.4 | 28,070 | 5.54 | Sept. | 33.80 | 133.63 | 0.98 | – |
| Cuba | 0.783 | 78.8 | 14.3 | 11.8 | 8621 | 5.33 | July | 65.28 | 241.8 | 1.17 | – |
| Cyprus | 0.887 | 81.0 | 15.2 | 12.2 | 38,207 | 3.40 | May | 75.00 | 850.6 | 2.39 | – |
| Ecuador | 0.759 | 77.0 | 14.6 | 8.9 | 11,044 | 1.39 | May | 75.46 | 94.51 | 4.52 | −0.18 |
| El Salvador | 0.673 | 73.3 | 11.7 | 6.9 | 8359 | 1.20 | Aug. | 32.41 | 42.27 | 1.82 | – |
| Georgia | 0.812 | 73.8 | 15.3 | 13.1 | 14,429 | 2.89 | Oct. | 47.22 | 394.85 | 8.15 | – |
| Germany | 0.947 | 81.3 | 17.0 | 14.2 | 55,314 | 8.00 | May | 75.00 | 224.56 | 2.79 | 0.13 |
| Honduras | 0.634 | 75.3 | 10.1 | 6.6 | 5308 | 0.64 | April | 82.41 | 60.22 | 1.35 | – |
| India | 0.645 | 69.7 | 12.2 | 6.5 | 6,681 | 0.53 | May | 73.61 | 266.28 | 2.38 | 0.73 |
| Indonesia | 0.718 | 71.7 | 13.6 | 8.2 | 11,459 | 1.04 | April | 71.76 | 18.25 | 0.5 | 0.5 |
| Iran | 0.783 | 76.7 | 14.8 | 10.3 | 12,447 | 1.56 | Aug. | 68.06 | 356.33 | 3.69 | 0.38 |
| Israel | 0.919 | 83.0 | 16.2 | 13.0 | 40,187 | 2.98 | May | 52.78 | 7.8 | 0.2 | −0.40 |
| Japan | 0.919 | 84.6 | 15.2 | 12.9 | 42,932 | 12.98 | May | 49.07 | 39.94 | 0.45 | 0.19 |
| Lebanon | 0.744 | 78.9 | 11.3 | 8.7 | 14,655 | 2.73 | Dec. | 31.48 | 216.93 | 1.25 | – |
| Lithuania | 0.882 | 75.9 | 16.6 | 13.1 | 35,799 | 6.43 | March | 66.67 | 194.12 | 4.04 | – |
| Mexico | 0.779 | 75.1 | 14.8 | 8.8 | 19,160 | 0.98 | Feb. | 71.76 | 107.44 | 9.68 | −0.35 |
| New Zealand | 0.931 | 82.3 | 18.8 | 12.8 | 40,799 | 2.57 | July | 22.22 | 0.47 | 0 | – |
| Pakistan | 0.557 | 67.3 | 8.3 | 5.2 | 5005 | 0.63 | July | 63.89 | 4.3 | 0.12 | – |
| Peru | 0.777 | 76.7 | 15.0 | 9.7 | 12,252 | 1.59 | Feb. | 80.56 | 189.71 | 16.68 | −0.34 |
| Philippines | 0.718 | 71.2 | 13.1 | 9.4 | 9778 | 0.99 | July | 71.76 | 51.56 | 0.98 | – |
| Poland | 0.880 | 78.7 | 16.3 | 12.5 | 31,623 | 6.54 | March | 73.15 | 261.22 | 6.07 | −0.32 |
| Portugal | 0.864 | 82.1 | 16.5 | 9.3 | 33,967 | 3.45 | Feb. | 76.85 | 1169.09 | 28.61 | 0.10 |
| Serbia | 0.806 | 76.0 | 14.7 | 11.2 | 17,192 | 5.61 | Feb. | 56.48 | 246.9 | 2.77 | – |
| Singapore | 0.938 | 83.6 | 16.4 | 11.6 | 88,155 | 2.49 | March | 50.93 | 1.81 | 0 | 0.36 |
| Slovakia | 0.860 | 77.5 | 14.5 | 12.7 | 32,113 | 5.70 | April | 74.07 | 492.18 | 12.51 | −0.36 |
| Turkey | 0.820 | 77.7 | 16.6 | 8.1 | 27,701 | 2.85 | May | 87.04 | 433.38 | 4.19 | 0.29 |
| Ukraine | 0.779 | 72.1 | 15.1 | 11.4 | 13,216 | 7.46 | Sept. | 50.93 | 49.81 | 1.48 | −0.44 |
| United Kingd. | 0.932 | 81.3 | 17.5 | 13.2 | 46,071 | 2.46 | Dec. | 46.76 | 634.01 | 1.79 | −0.21 |
| Vietnam | 0.704 | 75.4 | 12.7 | 8.3 | 7433 | 2.60 | May | 69.91 | 0.16 | 0 | 0.39 |

Notes: Higher Scores on HDI (Human Development Index) represent a higher level of development; scores were extracted from https://hdr.undp.org/data-center/documentation-and-downloads[55]. Data on hospital beds per thousand were extracted from https://ourworldindata.org/grapher/hospital-beds-per-1000-people[75]. Higher scores on government stringency represent more stringent and severe COVID-19 restrictions; each country's score represents the average stringency level for the month where data collection started; data on stringency level were extracted from https://covidtracker.bsg.ox.ac.uk/stringency-map[76]. The scores for daily new COVID-19 cases and deaths represent a 7-days rolling average per million people, assessed at the start of data collection; data were extracted from https://ourworldindata.org/explorers/coronavirus-data-explorer?uniformYAxis=0&pickerSort=asc&pickerMetric=location&Metric=Cases+and+deaths&Interval=7-day+rolling+average&Relative+to+Population=true&Color+by+test+positivity=false[77]. Higher scores on cultural tightness represent more tight cultures; scores for cultural tightness were extracted from https://www.thelancet.com/action/showPdf?pii=S2542-5196%2820%2930301-6[78].

and Chile. The raw dataset is made available at https://osf.io/kws9x/files/[43]. The final dataset consisted of $N = 12,758$ individual responses ($M_{age} = 26.9$; 67% women) from adult participants, collected over a period of approximately one year (from February 2021 to December 2021) across 34 different nations (Table 2). Participation to this research was voluntary and no monetary compensation was given to respondents for completing this research.

**Measures.** In terms of socio-demographic background variables, respondents were asked to indicate their age, educational degree and gender, with the following answer options for the latter: man, woman, none of the above, and non-binary. For descriptive reasons, respondents were further asked to report their nationality and their ethnicity, in the format of an open-ended question. No exclusions or additional analyses were made on the basis of respondents' gender identification, nationality or ethnicity; and none of the socio-demographic background questions was mandatory to answer.

To assess concepts associated with COVID-19 and its impact both already established and newly developed scales were administered. Please note that the entire survey also assessed concepts that go beyond the scope of the present research. Examples of the concepts are COVID-19 threat perception, national identity, hope, mindfulness, and religiousness. The survey was first developed in English and then translated and adapted to the local contexts (where necessary) by using the committee approach[44,45]. The survey was anonymous and completed on a voluntary basis (within 25 min approximately). The measures analyzed for the purpose of this study were: trust in the government, fear of COVID-19, empathic prosocial concern, and support for COVID-19 containment behaviors. The entire survey containing all scales is available at https://osf.io/kws9x/[46].

**Table 2 Participating countries and their descriptive statistics.**

| Country | N | Gender (% of women; men; & other/non-binary) | Mean age (years) | Individual-level trust in government; M(SD) (1 = low; 5 = high) | Fear of COVID-19; M(SD) (1 = low; 5 = high) | Empathic prosocial concern; M(SD) (1 = low; 7 = high) | Support for COVID-19 containment Beh. M(SD) (1 = low; 5 = high) | Country-level trust in government; % with high trust |
|---|---|---|---|---|---|---|---|---|
| Australia | 301 | 76% f; 23% m; 1% o | 23.3 | 2.72 (0.81) | 2.06 (0.80) | 5.51 (1.31) | 3.27 (0.77) | 30.3 |
| Bangladesh | 714 | 53% f; 47% m | 22.7 | 3.04 (0.94) | 3.21 (0.79) | 5.29 (1.23) | 4.30 (0.73) | 81.1 |
| Brazil | 502 | 73% f; 26% m; 1% o | 33.6 | 1.62 (0.81) | 2.97 (0.87) | 6.00 (1.12) | 4.30 (0.56) | 22.4 |
| Bulgaria | 294 | 63% f; 37% m | 38.3 | 2.49 (1.08) | 2.34 (0.88) | 5.13 (1.12) | 3.46 (1.00) | 19.9 |
| Colombia | 339 | 62% f; 38% m | 24.2 | 2.62 (0.97) | 2.62 (0.81) | 5.66 (1.10) | 3.94 (0.62) | 11.9 |
| Croatia | 540 | 63% f; 37% m | 25.4 | 2.23 (0.86) | 1.83 (0.71) | 4.82 (1.41) | 2.47 (0.75) | 9.6 |
| Cuba | 301 | 52% f; 48% m | 24.1 | 2.60 (1.13) | 2.67 (0.98) | 4.42 (1.73) | 3.58 (1.03) | – |
| Cyprus | 173 | 66% f; 34% m | 37.1 | 2.33 (0.96) | 2.30 (0.77) | 5.77 (1.00) | 3.59 (0.77) | 33.4 |
| Ecuador | 199 | 67% f; 33% m | 27.5 | 2.51 (0.98) | 2.82 (0.96) | 5.21 (1.26) | 4.00 (0.66) | 31.8 |
| El Salvador | 261 | 58% f; 42% m | 27.7 | 2.12 (0.99) | 2.69 (0.90) | 5.30 (1.32) | 4.03 (0.56) | – |
| Georgia | 179 | 83% f; 17% m | 25.6 | 2.26 (1.00) | 2.49 (0.85) | 5.50 (1.20) | 3.73 (0.75) | 37.4 |
| Germany | 930 | 77% f; 22% m; 1% o | 31.1 | 2.79 (0.87) | 1.86 (0.70) | 5.31 (1.26) | 3.00 (0.76) | 44.2 |
| Honduras | 718 | 62%f; 38% m | 26.0 | 1.52 (0.88) | 2.78 (1.03) | 5.05 (1.70) | 4.09 (0.84) | – |
| India | 251 | 36% f; 64% m | 21.2 | 3.42 (0.89) | 3.04 (0.84) | 5.65 (0.80) | 4.53 (0.52) | – |
| Indonesia | 196 | 76% f; 24% m | 23.1 | 2.88 (0.75) | 2.72 (0.75) | 5.59 (1.03) | 4.13 (0.49) | 78.8 |
| Iran | 245 | 61% f; 39% m | 31.5 | 2.15 (0.98) | 2.46 (0.86) | 5.60 (1.11) | 4.12 (0.72) | 51.7 |
| Israel | 201 | 50% f; 50% m | 30.1 | 2.45 (0.98) | 2.00 (0.88) | 5.32 (1.55) | 3.66 (0.87) | – |
| Japan | 874 | 52% f; 48% m | 36.3 | 2.28 (0.83) | 2.84 (0.80) | 4.28 (1.39) | 3.79 (0.60) | 39.9 |
| Lebanon | 208 | 73% f; 27% m | 25.4 | 1.24 (0.56) | 2.27 (0.81) | 5.49 (1.30) | 3.50 (0.79) | 19.8 |
| Lithuania | 342 | 81% f; 19% m | 32.0 | 2.67 (0.84) | 2.19 (0.79) | 4.62 (1.29) | 3.59 (0.77) | 39.0 |
| Mexico | 205 | 87% f; 13% m | 24.1 | 1.59 (0.74) | 2.78 (0.99) | 5.81 (1.27) | 4.30 (0.48) | 17.4 |
| New Zealand | 188 | 80% f; 19% m; 2% o | 19.6 | 3.14 (0.81) | 2.13 (0.78) | 5.32 (1.33) | 2.82 (0.81) | 50.0 |
| Pakistan | 218 | 93% f; 7% m | 21.3 | 3.12 (0.85) | 2.62 (0.83) | 4.82 (1.29) | 3.08 (1.22) | 62.3 |
| Peru | 705 | 72% f; 28% m | 24.2 | 1.99 (0.88) | 3.03 (1.04) | 5.70 (1.38) | 4.32 (0.73) | 10.6 |
| Philippines | 475 | 64% f; 36% m | 21.4 | 2.39 (1.00) | 3.31 (0.84) | 6.00 (1.01) | 4.46 (0.43) | 81.6 |
| Poland | 923 | 59% f; 41% m | 27.4 | 1.57 (0.79) | 2.11 (0.78) | 4.74 (1.40) | 3.38 (0.88) | 23.1 |
| Portugal | 185 | 69% f; 30% m; 1% o | 39.2 | 2.69 (0.99) | 2.51 (0.86) | 5.98 (1.02) | 4.35 (0.54) | 34.4 |
| Serbia | 569 | 85% f; 15% m | 22.6 | 2.07 (0.83) | 2.12 (0.74) | 5.50 (1.29) | 3.11 (0.83) | 28.7 |
| Singapore | 347 | 71% f; 29% m | 25.4 | 3.49 (0.68) | 2.10 (0.69) | 5.08 (1.23) | 3.32 (0.58) | 80.6 |
| Slovakia | 152 | 63% f; 37% m | 18.4 | 2.28 (0.82) | 2.26 (0.74) | 4.92 (1.35) | 3.42 (0.83) | 30.4 |
| Turkey | 405 | 80% f; 20% m | 21.9 | 1.89 (0.98) | 2.64 (0.96) | 5.36 (1.09) | 4.21 (0.61) | 68.8 |
| Ukraine | 167 | 63%f; 37% m | 20.2 | 2.40 (1.00) | 2.07 (0.88) | 4.43 (1.65) | 2.92 (0.89) | 18.9 |
| United Kingd. | 152 | 77% f, 18% m, 5% o | 24.3 | 1.96 (0.85) | 2.33 (0.81) | 5.69 (1.23) | 3.08 (0.92) | 29.3 |
| Vietnam | 286 | 67% f; 33% m | 23.2 | 3.60 (0.71) | 3.04 (0.85) | 4.68 (1.21) | 4.28 (0.59) | 92.9 |

Trust in the government (individual level) was assessed with four items adapted from Kerr and colleagues[47]. Respondents were asked to rate items (e.g., My government is there for me when I need it) on a five-point Likert scale ranging (from *1 = strongly disagree* to *5 = strongly agree)*. The internal consistency was calculated as Cronbach's alpha = 0.95 for the total sample, and ranged between 0.85 and 0.97 across the different countries.

The country-level scores for trust in the government were extracted from the most recent wave of the World Values Survey[48]. The World Values Survey asks nationally representative respondents to indicate their level of confidence in their government with answer options ranging from *none at all, not very much,* to *quite a lot,* and *a great deal* (additionally there were the answer options *don't know* and *no answer*). The data used for the present analyses were all collected in 2017 and 2020. We used the sum of the percentage of respondents who reported to have *a great deal* and *quite a lot* of confidence in their governments as an indicator of the general level of governmental trust in the different study contexts, (for a similar procedure see ref. [49], for summary of our results see Table 2).

The three items from the research by Pfatteicher and colleagues[19] were used to assess people's empathic concern for others in times of the COVID-19 pandemic. Using a Likert scale ranging from *1 = strongly disagree* to *7 = strongly agree*, respondents were asked to indicate how much they agree, for instance, with the following statement: I am very concerned about those most vulnerable to COVID-19. Cronbach's alpha for the overall sample was 0.91, and ranged between 0.63 and 0.97 across the countries.

The seven-item fear of COVID-19 scale[50] was used to assess respondents' fear. On a five-point Likert scale ranging from *1 = strongly disagree* to *5 = strongly agree* respondents were asked to indicate how much they agree with, such as for instance: I am most afraid of COVID-19. Cronbach's alpha was. 90 for the total sample, and ranged between 0.83 and 0.93 across the different countries.

Support for COVID-19 containment behaviors was used to assess participants' engagement in COVID-19 containment behaviors, an extended version of the measure developed by Tepe and Karakulak[51] was used. With 13 items, respondents were asked to report how important it is to engage in a particular COVID-19 preventive behavior (such as wearing a mask, not going outside, trying to stay at home, or frequently washing hands) in the examined period. Response options ranged from *1 = not at all important* to *5 = very important*. Internal consistencies were calculated as Cronbach's alpha = 0.94 for the total sample, and ranged between 0.81 and 0.95 across the different countries.

**Data analysis**. The hypotheses of the present research were tested with two separate multi-level regression analyses using the maximum likelihood estimator. First a stepwise linear regression nested within the study countries was carried out to test the proposed associations on an individual-level; and second a mixed-level stepwise linear regression was carried out to test the proposed interaction with country-level scores of trust in the government. The regression analyses were carried out across four steps. First, a null model without any predictors, and only the criterion variable, was carried out. Second, a fixed-effects model that specified random intercepts and fixed slopes for the main effects of the predictor variables was calculated. Third, a random slopes model testing the predictors' main effects was carried out; and finally, the hypothesized two-way interactions were added to the random slopes regression model. All predictor variables entered into the model were group-mean centered, except for the

country-level score of trust in the government which was grand-mean centered. All analyses (from the second step and onwards) were conducted by entering covariate effects of respondents' age and their gender, and country-level scores of HDI, number of hospital beds per 1000, month of data collection, government stringency level, and the number of new daily COVID-19 cases and deaths by the time of data collection. Due to the large sample size of the present study, the data distribution was not formally tested for normality, but assumed to be normal. The descriptive statistics, histogram and Q-Q- plot of the dependent variable are presented in the Supplementary Information under Supplementary Note 2, Supplementary Table 2, Supplementary Figs. 1 and 2.

**Exploratory analysis**. We conducted additional analyses to examine (1) whether our findings remain robust when generalized trust is added as another covariate to the analyses, and (2) whether the proposed moderation effects can also be obtained with generalized (instead of governmental) trust. The scores for individual-level generalized trust were obtained by averaging the scores that respondents provided to the following two questions: "In general, most people in our community can be trusted", and "Most people in our community are fair and do not take advantage of you" (*1 = totally disagree*, *5 = totally agree*, *r* = 0.78). The scores for country-level generalized trust were extracted from the most recent wave of the World Values Survey[48].

**Reporting summary**. Further information on research design is available in the Nature Portfolio Reporting Summary linked to this article.

## Results

We first tested our hypotheses at the individual level with the pooled individual-level data nested within countries, and then ran an additional test with country-level scores of trust in the government extracted from the World Values Survey[48] as a moderating factor. Analyses were conducted with the program jamovi 2.0[52].

**Individual level analyses**. The overall dataset comprises complete responses from 12,758 individuals living in 34 countries. Table 2 illustrates the number of responses, information about participants' age and gender, and the descriptive statistics of the study variables per country. Table 3 shows the Pearson product-moment correlations for the study variables based on the pooled dataset.

We conducted a multi-level linear regression model to estimate individual-level differences, nested within 34 countries. Results from this analysis using the maximum likelihood estimator are summarized in Table 4. First, a null model without any predictors, and the support for COVID-19 containment behaviors as the only criterion variable, was implemented[53]. In this model the ICC(1) for the criterion variable was 0.34, suggesting that 34% of the variance in supporting COVID-19 containment behaviors existed between the different countries, which justifies the use of a mixed-level approach that takes this between-level variance into account. In the second step, we ran a fixed-effects model that specified random intercepts and fixed slopes for the main effects of the predictor variables: individuals' trust in their governments, their empathic prosocial concern, and their fear of COVID-19. We used group-mean centering for these scores as grand-mean centering creates inappropriate level-1 estimators that reflect a mixture of within and between group variations[54]. Additionally, to account for possible socio-demographic, country- and pandemic-specific effects, we entered respondents' age and

their gender, and country-level scores of the Human Development Index (HDI, a composite score reflecting the level of a country's overall development in the domains of economy, education, and health)[55], number of hospital beds per 1000, month of data collection, government stringency level, and the number of new daily COVID-19 cases and deaths by the time of data collection as covariates to the analyses. The analyses suggested that a higher level of trust in the government ($B$(SE) = 0.05 (0.01), 95% $CI$ = [0.04, 0.06], $t$(12705.1) = 7.60, $p < 0.001$, Cohen's $f = 0.07$), more empathic prosocial concern ($B$(SE) = 0.15 (0.05), 95% $CI$ = [0.14, 0.16], $t$(12705.3) = 32.04, $p < 0.001$, Cohen's $f = 0.28$), and stronger fear of COVID-19 ($B$(SE) = 0.29 (0.01), 95% $CI$ = [0.27, 0.30], $t$(12705.3) = 39.94, $p < 0.001$, Cohen's $f = 0.35$) were all significantly associated with stronger support for COVID-19 containment behaviors.

In the next step (step 3), we repeated the same analysis using the random slopes model whereby the slopes (the strength of the main effects) of trust in the government, empathic concern and fear of COVID-19 were allowed to vary across clusters. This is because the different preconditions that exist across countries (i.e., number of daily infections, mortality rates, lockdown regulations, etc.) may affect how strongly support for COVID-19 containment behaviors is associated with trust in the government, empathic prosocial concern, and fear of COVID-19. Results again confirmed that all three predictor variables were significantly and positively associated with support for COVID-19 containment behaviors, ($B$(SE) = 0.05 (0.01), 95% $CI$ = [0.03, 0.08], $t$(34.5) = 4.11, $p < 0.001$, and Cohen's $f = 0.66$ for trust in the government; $B$(SE) = 0.14 (0.01), 95% $CI$ = [0.12, 0.15], $t$(30.4) = 18.08, $p < 0.001$, and Cohen's $f = 3.17$ for empathic concern; and $B$(SE) = 0.30 (0.02), 95% $CI$ = [0.26, 0.35], $t$(33.1) = 12.71, $p < 0.001$, and Cohen's $f = 2.14$ for fear of COVID-19).

In step four, the hypothesized two-way interactions were added to the regression model. First, we compared the deviance scores (calculated as -2*loglikelihood) of the random coefficient model with the main effects only (step 3; null model) with the random coefficients model that also included the two-way interactions (step 4; alternative model) to test whether one or the other model explained significantly more variance[56]. The deviance score of the null model was 24,864. The deviance score of the alternative model including the interactions was smaller with 24,789. Results of the Chi-square test suggest that the difference between these two models was significant [$\chi^2$ (3, $N = 12,758$) = 75, $p < 0.001$], indicating that the model including the interactions was significantly better at explaining the variance in supporting COVID-19 containment behaviors. The results for this regression model show that individuals' trust in their government significantly interacted with both the individuals' empathic prosocial concern ($B$(SE) = 0.01 (0.005), 95% $CI$ = [0.003, 0.02], $t$(12352.8) = 2.56, $p = 0.011$, Cohen's $f = 0.02$), and their fear of COVID-19 ($B$(SE) = -0.02 (0.01), 95% $CI$ = [-0.03, -0.005], $t$(12435.6) = -2.64, $p = 0.008$, Cohen's $f = 0.02$). We decomposed the significant interactions and examined the associations between support for COVID-19 with empathic prosocial concern and fear of COVID-19 under low (-1$SD$), medium (mean) and high (+1$SD$) levels of trust in the government. Empathic prosocial concern was significantly associated with the support for COVID-19 containment behaviors across all three levels of trust in the government. As illustrated in Fig. 1, the association of empathic prosocial concern and support for COVID-19 containment measures was strongest when trust in the government was high ($\beta = 0.14$, $t$(59.0) = 15.8, $p < 0.001$, 95% $CI$ [0.12, 0.16], Cohen's $f = 2.01$), and weakest when trust in the government was low ($\beta = 0.12$, $t$(47.5) = 14.1, $p < 0.001$, 95% $CI$ [0.10, 0.14], Cohen's $f = 1.99$), which supports Hypothesis 1a. Fear of

**Table 3 Pearson product-moment correlations of the study variables.**

| | 1. | 2. | 3. | 4. |
|---|---|---|---|---|
| 1. Support for COVID-19 Containment Behaviors | 1 | 0.36*** | 0.54*** | 0.05*** |
| 2. Empathic Prosocial Concern | | 1 | 0.29*** | 0.04*** |
| 3. Fear of COVID-19 | | | 1 | 0.07*** |
| 4. Trust in the Government (Individual-level) | | | | 1 |

$N = 12,758$ independent responses.
***$p < 0.001$.

COVID-19 was also significantly associated with support for COVID-19 containment behaviors at all levels of trust in the government. Conversely to empathic concern, the association of fear of COVID-19 and support for COVID-19 containment behaviors was strongest when trust in the government was low ($\beta = 0.32$, $t$(38.1) = 13.0, $p < 0.001$, 95% $CI$ [0.27, 0.37], Cohen's $f = 2.04$), and weakest when trust in the government was high ($\beta = 0.29$, $t$(38.1) = 11.6, $p < 0.001$, 95% $CI$ [0.24, 0.34], Cohen's $f = 1.83$), which supports Hypothesis 2a (Fig. 2).

**Mixed-level analyses with country-level scores of trust in the government**. We repeated the same regression analysis as described above, using country-level scores for trust in the government (see Table 5). The mixed-level linear regression analysis comprised the data of 11,026 responses from 29 countries (the data from five countries that did not participate in the World Values Survey had to be excluded from the analyses. These countries were Honduras, India, Israel, El Salvador, and Cuba).

The null model without any predictor variables was the same as in the previous analysis and showed that 34% of the variance in the support for COVID-19 containment behaviors existed between the different countries, ICC(1) = 0.34. In the second step, we ran a fixed-effects model and entered the main effects of the above-described covariates and the predictor variables into our model: group-mean centered scores of empathic prosocial concern and fear of COVID-19 were used, while the country-level trust in the government score was grand-mean centered. The results show that both empathic prosocial concern ($B$(SE) = 0.15 (0.01), 95% $CI$ = [0.14, 0.16], $t$(10978.2) = 29.67, $p < 0.001$, Cohen's $f = 0.28$) and fear of COVID-19 ($B$(SE) = 0.29 (0.01), 95% $CI$ = [0.28, 0.31], $t$(10978.2) = 37.17, $p < 0.001$, Cohen's $f = 0.35$) were significantly associated with supporting COVID-19 containment behaviors, whereas no evidence was found for an association with country-level trust in the government ($B$(SE) = 0.004 (0.004), 95% $CI$ = [-0.03, 0.01], $t$(29.1) = 1.20, $p = 0.24$, Cohen's $f = 0.22$).

In step three, we assumed a random slopes model, whereby the effects of governmental trust, empathic prosocial concern and fear of COVID-19 could freely vary across countries. Again, supporting COVID-19 containment behaviors was significantly associated with both empathic prosocial concern ($B$(SE) = 0.14 (0.01), 95% $CI$ = [0.13, 0.16], $t$(24.5) = 17.60, $p < 0.001$, Cohen's $f = 3.41$), and fear of COVID-19 ($B$(SE) = 0.31 (0.03), 95% $CI$ = [0.25, 0.36], $t$(28.5) = 11.06, $p < 0.001$, Cohen's $f = 2.00$), while no evidence was found for a significant association with country-level trust in the government ($B$(SE) = 0.002 (0.004), 95% $CI$ = [-0.007, 0.01], $t$(17.5) = 0.40, $p = 0.69$, Cohen's $f = 0.09$). In the final step, we tested the proposed moderator effect of trust in the government as a context variable in accordance with our hypotheses and added the two-way interactions to the regression model. Again, the deviance scores

**Table 4 Mixed-level regression results with individual-level scores for trust in the government.**

| | Model | | | |
|---|---|---|---|---|
| | Step 1<br>Null model<br>B (SE) | Step 2<br>Random intercept and fixed slopes<br>B (SE) | Step 3<br>Random intercept and random slopes<br>B (SE) | Step 4<br>Two-way interactions<br>B (SE) |
| Intercept | 3.71*** (0.09) | 3.78*** (0.09) | 3.78*** (0.09) | 3.80*** (0.09) |
| Main effects | | | | |
| Trust in government (TG) | | 0.05*** (0.01) | 0.05*** (0.01) | 0.05*** (0.01) |
| Empathic concern (EC) | | 0.15*** (0.005) | 0.14*** (0.01) | 0.13*** (0.01) |
| Fear of COVID-19 (FoC) | | 0.29*** (0.01) | 0.30*** (0.02) | 0.31*** (0.02) |
| Interactions | | | | |
| TG × EC | | | | 0.01* (0.005) |
| TG × FoC | | | | −0.02** (0.01) |
| EC × FoC | | | | −0.04*** (0.005) |
| Variance Components | | | | |
| Within-country variance | 0.553 | 0.419 | 0.403 | 0.401 |
| Between-country variance | 0.283 | 0.127 | 0.136 | 0.136 |
| TG between-country var. | | | 0.004 | 0.004 |
| EC between-country var. | | | 0.001 | 0.001 |
| FoC between-country var. | | | 0.017 | 0.017 |
| Additional Information | | | | |
| −2 *log likelihood (FIML) | 28,772 | 25,221 | 24,864 | 24,789 |
| $R^2$ marginal | 0.000 | 0.37 | 0.33 | 0.33 |

$N = 12,758$ independent responses nested in 34 countries. Coefficients presented for main effects and interactions represent the unstandardized regression weights (B); the value in brackets refers to the Standard Error (SE). All analyses from step 2 onwards were performed by entering covariate effects of gender, age, HDI (Human Development Index), hospital beds per 1000, month of data collection, government stringency level, and the number of new daily COVID-19 cases and deaths by the time of data collection. For reasons of simplicity, the covariate effects are not displayed in the table. They can be obtained from the analysis documentation at https://osf.io/kws9x/files[43].
*TG* Trust in Government, *EC* Empathic Concern, *FoC* Fear of COVID-19.
*$p < 0.05$; **$p < 0.01$; ***$p < 0.001$.

of the random coefficients model at step 3 (=21,254) was compared with that of the model including the interactions at step 4 ( = 21,191)[56]. Results of the Chi-square test showed that the latter model explained significantly more variance, $\chi^2$ (3, $N = 11,026$) = 63, $p < 0.001$. There was no significant interaction between country-level scores of trust in government and empathic prosocial concern in the regression analysis ($B$(SE) = −0.0003 (0.0003), 95% $CI = [−0.0009, 0.0004]$, $t(24.8) = −0.82$, $p = 0.42$, Cohen's $f = 0.16$). Hence, we found no credible evidence to support Hypothesis 1b. For fear of COVID-19, a significant interaction with country-level trust in the government was found ($B$(SE) = −0.002 (0.0008), 95% $CI = [−0.004, −0.0007]$, $t(25.9) = −2.92$, $p = 0.007$, Cohen's $f = 0.52$). Examination of the simple effects showed that the effect of fear of COVID-19 was significantly associated with supporting COVID-19 containment behaviors across low (−1 SD), medium (mean) and high (+1 SD) levels of trust in the government. However, the association between fear of COVID-19 and support for COVID-19 containment behaviors was strongest in contexts where trust in the government is generally low ($\beta = 0.37$, $t(37.3) = 11.26$, $p < 0.001$, 95% $CI$ [0.30, 0.43], Cohen's $f = 1.79$), and weakest in contexts where trust in the government is generally high ($\beta = 0.26$, $t(35.9) = 8.06$, $p < 0.001$, 95% $CI$ [0.19, 0.32], Cohen's $f = 1.30$), which supports Hypothesis 2b (Fig. 3).

**Exploratory analysis**. We examined whether adding individual-level scores of generalized trust as another covariate into the mixed-linear regression model (at step 4) will lead to substantial changes in the obtained results. Results revealed that individuals' trust in their government significantly interacted with both the individuals' empathic prosocial concern ($B$(SE) = 0.013 (0.005), 95% $CI$ [0.003, 0.02], $t(12080.2) = 2.65$, $p = 0.008$, Cohen's $f = 0.02$), and their fear of COVID-19 ($B$(SE) = −0.02

(0.007), 95% $CI$ [−0.003, −0.03], $t(12130.0) = −2.42$, $p = 0.015$, Cohen's $f = 0.02$); both in the proposed direction. Moreover, the same analyses were repeated by controlling for effects of country-level generalized trust. Again, the original analysis results did not change, and analyses supported that trust in the government moderated the association between fear of COVID-19 and support for COVID-19 containment behaviors ($B$(SE) = −0.002 (0.001), 95% $CI$ [−0.004, −0.0001], $t(26.8) = −2.09$, $p = 0.047$, Cohen's $f = 0.34$) in the expected direction; while we found no evidence for a significant moderation for the association between empathic concern and support for COVID-19 containment behaviors ($B$(SE) = −0.0003 (0.0003), 95% $CI$ [−0.0008, 0.0003], $t(30.2) = −0.97$, $p = 0.34$, Cohen's $f = 0.17$).

We further tested the hypotheses of the present research on the basis of generalized trust (instead of governmental trust) as a moderator of self- versus other-oriented motives. The analysis results for this regression model did not provide credible evidence for a significant interaction; neither between generalized trust and empathic concern ($B$(SE) = −0.0004 (0.004), 95% $CI$ [−0.009, 0.009], $t(12389.5) = 0.07$, $p = 0.95$, Cohen's $f = 0.001$), nor between generalized trust and fear of COVID-19 ($B$(SE) = −0.0003 (0.0003), 95% $CI$ [−0.02, 0.005], $t(12394.3) = −1.28$, $p = 0.20$, Cohen's $f = 0.01$). Second, the same analyses were repeated on the country-level with generalized trust scores extracted from the World Values Survey. Again, the results obtained from the regression analysis did not provide credible support for the proposed interactions (H1b and H2b). Both the interaction between generalized trust and empathic concern ($B$(SE) = −0.0007 (0.0006), 95% $CI$ [−0.002, 0.0004], $t(30.8) = −1.26$, $p = 0.22$, Cohen's $f = 0.23$) and between generalized trust and fear of COVID-19 ($B$(SE) = 0.003 (0.002), 95% $CI$ [−0.0003, 0.006], $t(27.7) = 1.77$, $p = 0.09$, Cohen's $f = 0.33$) were not found as significant.

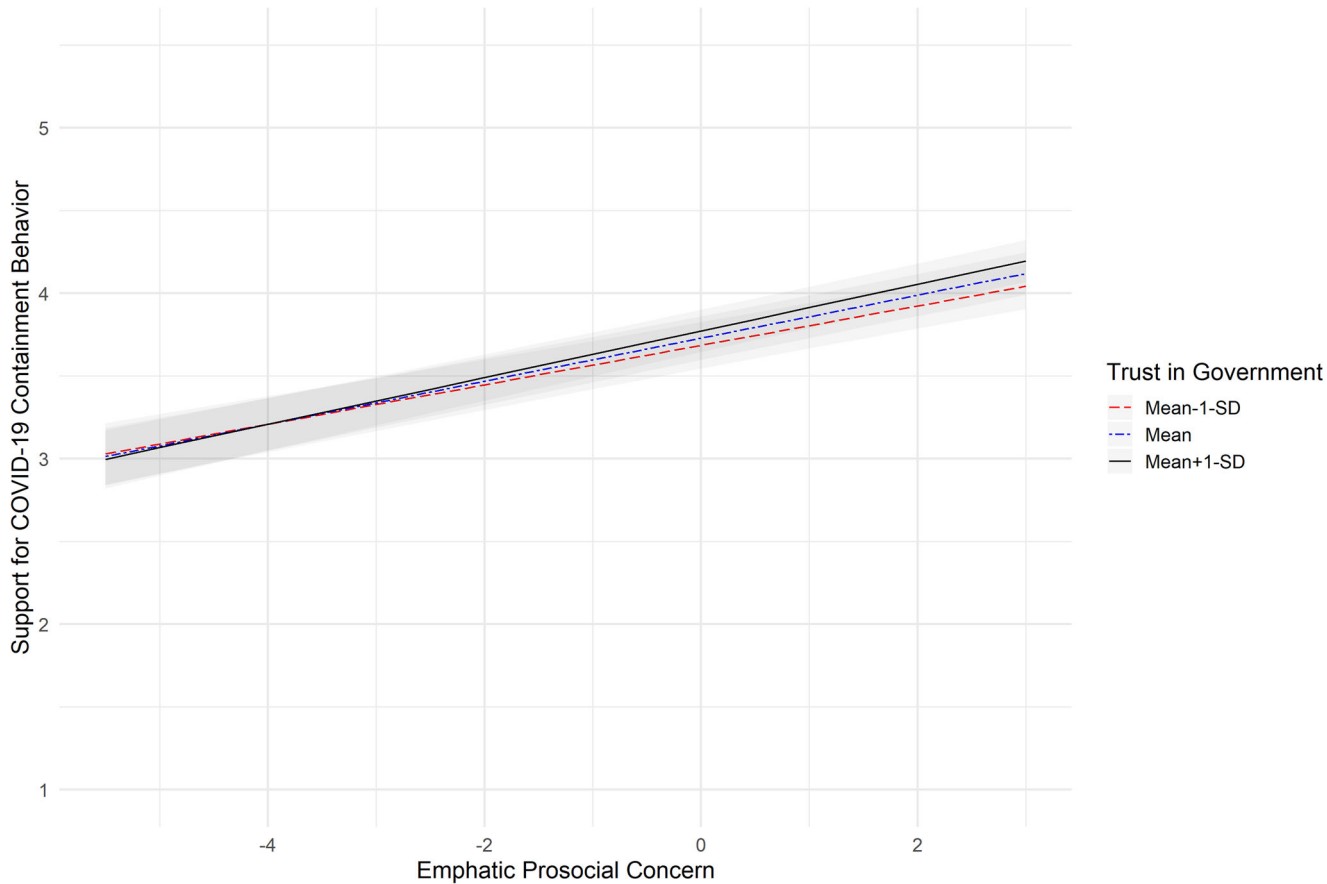

**Fig. 1 Association between empathic prosocial concern and support for COVID-19 containment behaviors under low, medium and high scores of trust in the government.** Notes: The graphic is based on $N = 12{,}758$ independent responses nested in 34 countries; the black line represents the strength of the association between empathic prosocial concern and support for COVID-19 containment behaviors under high levels of trust in the government (one standard deviation above the mean); the blue dashed line represents the strength of the association between empathic prosocial concern and support for COVID-19 containment behaviors under medium levels of trust in the government; the red dashed line represents the strength of the association between empathic prosocial concern and support for COVID-19 containment behaviors under low levels of trust in the government (one standard deviation below the mean). The gray shadowed parts represent the respective 95% Confidence Intervals; support for COVID-19 containment behaviors were assessed on a scale ranging from *1* to *5*, with higher scores representing higher support. The scores for empathic prosocial concern (x-axis) represent the group-mean centered scores.

Finally, we tested the robustness of our results when using a different methodological approach, namely a fixed effects regression model with cluster-robust standard errors. For this purpose, two additional analyses were performed: first, a simple OLS regression; and second, an OLS regression with the clustered standard errors correction. A table comparing the results obtained from these approaches, and a brief discussion concerning the differences in the results can be found in the Supplementary Information under Supplementary Note 1 and Supplementary Table 1.

**Discussion**

The present multinational study provides an individual- and country-level perspective on the role of governmental trust as a moderator for motivations to support COVID-19 containment behaviors. Ever since its emergence in 2020, the pandemic has been a major reason for worry and concern globally, so governments and local officials strove to efficiently implement varying strategies to combat the spread of the virus. So far, only few studies have examined how the motives of people to comply with public health guidelines may differ in accordance with various boundary conditions[37,57,58]. The present study tried to shed light

on the ways in which individuals with low versus high levels of trust in the government (or in context where trust is low versus high) respond to the COVID-19 pandemic, and how their motivations for supporting COVID-19 containment behaviors may differ accordingly.

First, we hypothesized that the positive association between empathic concern and the support of COVID-19 containment behaviors would be stronger at high compared to low levels of trust in the government. Our evidence supported this hypothesis at the individual-level but not the country-level. Namely, support for COVID-19 containment behaviors was more strongly associated with empathic concern when people reported high (compared to low) trust in their government. Second, and as hypothesized, we found that support for COVID-19 containment behaviors was more strongly associated with fear of COVID-19 when people reported low (compared to high) trust in their government, as well as within contexts where trust in the government is typically low (versus high).

In regard to empathic concern, it should be noted, that despite the significant interaction effect, the strength of the association between empathic concern and support for COVID-19 containment behaviors were almost identical under high (Cohen's $f = 2.01$) versus low trust in the government (Cohen's $f = 1.99$).

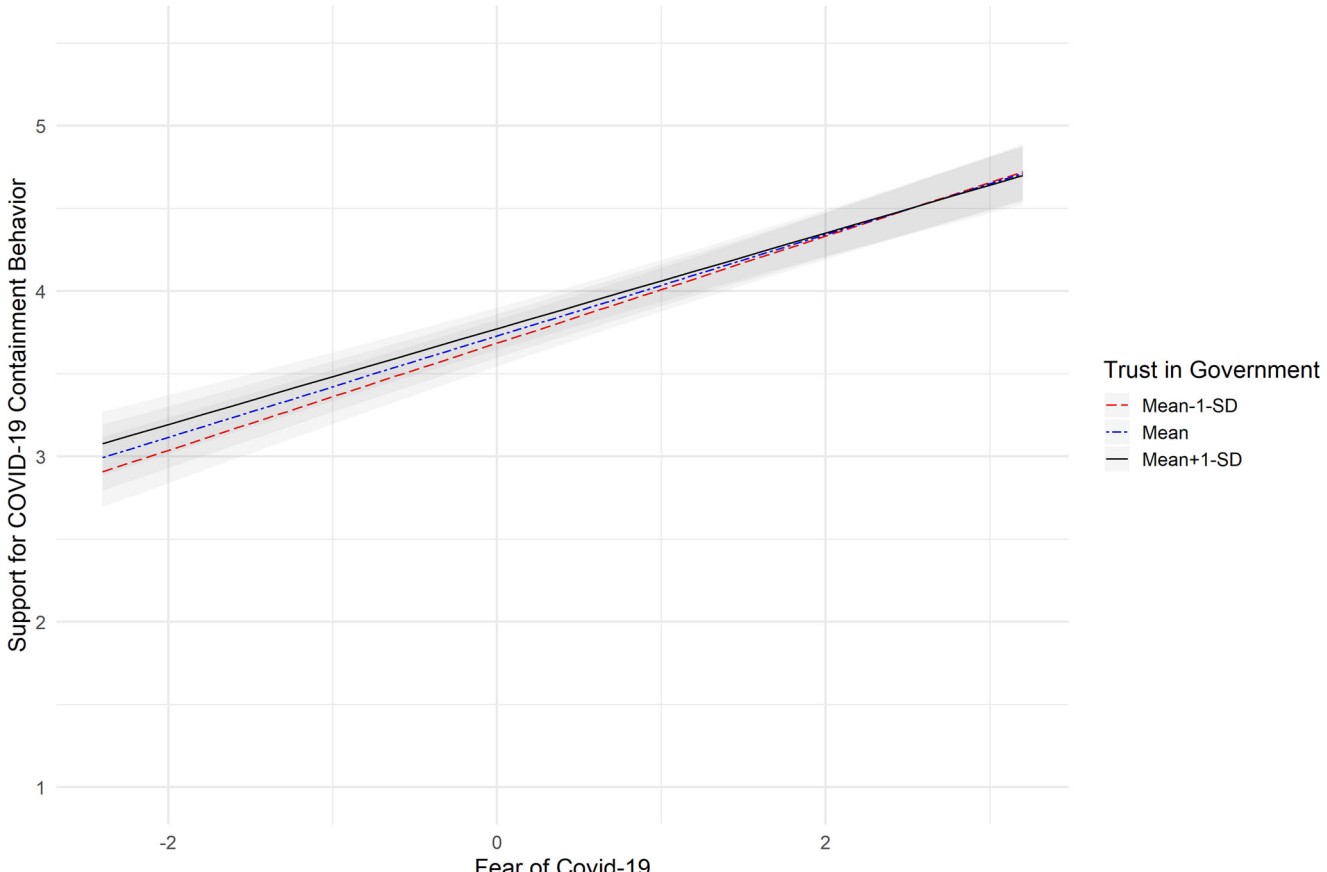

**Fig. 2 Association between fear of COVID-19 and support for COVID-19 containment behaviors under low, medium and high scores of trust in the government.** Notes: The graphic is based on $N = 12,758$ independent responses nested in 34 countries; the black line represents the strength of the association between fear of COVID-19 and support for COVID-19 containment behaviors under high levels of trust in the government (one standard deviation above the mean); the blue dashed line represents the strength of the association between fear of COVID-19 and support for COVID-19 containment behaviors under medium levels of trust in the government; the red dashed line represents the strength of the association between fear of COVID-19 and support for COVID-19 containment behaviors under low levels of trust in the government (one standard deviation below the mean). The gray shadowed parts represent the respective 95% Confidence Intervals; support for COVID-19 containment behaviors were assessed on a scale ranging from 1 to 5, with higher scores representing higher support. The scores for fear of COVID-19 (x-axis) represent the group-mean centered scores.

This aligns with the notion that people's empathic concern may similarly operate across various conditions. Thus, even though the association between empathy and support for large-scale cooperative action may become more pronounced when individuals perceive their governments as more (compared to less) trustworthy, such interplay with trust seems less salient for empathic concern (compared to fear of COVID-19). Overall, our results seem in line with past findings that empathy is consistently linked with prosocial behaviors[59,60], and that highly empathetic people usually show a higher sense of community and civic engagement. These people are more likely to take on the responsibility for the community's well-being[57], and could thus also be more prone to respond sensitively and favorably to issues of protecting citizens' health regardless of whether they find themselves in a context where trust in the government is (generally) low or high.

At both levels of analysis, our results support that fear of COVID-19 was related to preventive behaviors, especially when trust in the government was low. According to a number of authors[8,61,62], in times of epidemic outbreak, people, especially those at risk of infection, are extremely vulnerable, emotionally fragile, and feel ambivalent emotions. They feel fear and panic, but also skepticism and disregard. Under these circumstances, individuals have difficulty processing information and may think

of and behave for themselves as individuals rather than as connected to others. The same may also apply when people perceive their governments as not trustworthy, or when they are living in surroundings that are characterized by low levels of trust in the government. Under such conditions, individuals may feel a stronger need to curb their fear through personal protective behaviors, and thus their self-protection motives may play a stronger role for their support of COVID-19 containment behaviors. This is also in line with previous research showing that cooperation seems to rely on selfish (instead of altruistic) reasons, especially when deliberate cognitive processes are involved in the decision of whether to cooperate or not[17,18]. Yet, it should be noted that our results do not indicate that empathy would not operate in low trust settings, or that fear would not operate in high trust settings. Instead, our results support the view that both motivational pathways seem relevant for mobilizing cooperative action across individuals and contexts (i.e., as main effects were consistently recorded across various levels of governmental trust), but suggest that empathy-driven cooperation is likely to unfold optimally when trust in the government is perceived as high, while fear-driven cooperation is likely to become most pronounced in contexts where trust in the government is generally low, or when trust in the government is individually perceived as low.

**Table 5 Mixed-level regression results with country-level scores for trust in the government.**

| | Model | | | |
|---|---|---|---|---|
| | Step 1<br>Null model<br>B (SE) | Step 2<br>Random intercept and fixed slopes<br>B (SE) | Step 3<br>Random intercept and random slopes<br>B (SE) | Step 4<br>Two-way interactions<br>B (SE) |
| Intercept | 3.71*** (0.09) | 3.72*** (0.09) | 3.80*** (0.09) | 3.75*** (0.09) |
| Main effects | | | | |
| TG (Country-Level) | | 0.004 (0.004) | 0.002 (0.04) | 0.01 (0.004) |
| EC | | 0.15*** (0.01) | 0.14*** (0.01) | 0.13*** (0.01) |
| FoC | | 0.29*** (0.01) | 0.31*** (0.03) | 0.31*** (0.03) |
| Interactions | | | | |
| TG (Country-Lev.) ×EC | | | | −0.0003 (0.0003) |
| TG (Country-Lev.) ×FoC | | | | −0.002** (0.001) |
| EC × FoC | | | | −0.04*** (0.01) |
| Variance Components | | | | |
| Within-country variance | 0.553 | 0.410 | 0.396 | 0.394 |
| Between-country variance | 0.283 | 0.124 | 0.071 | 0.076 |
| TG between-country var. | | | 0.0003 | 0.0004 |
| EC between-country var. | | | 0.001 | 0.001 |
| FoC between-country var. | | | 0.020 | 0.018 |
| Additional Information | | | | |
| −2 *log likelihood (FIML) | 28,815 | 21,549 | 21,254 | 21,191 |
| $R^2$ marginal | 0.000 | 0.37 | 0.25 | 0.34 |

Notes: $N = 11{,}026$ independent responses nested in 29 countries. Coefficients presented for main effects and interactions represent the unstandardized regression weights (B); the value in brackets refers to the Standard Error (SE). All analyses from step 2 onwards were performed by entering covariate effects of gender, age, HDI (Human Development Index), hospital beds per 1000, month of data collection, government stringency level, and the number of new daily COVID-19 cases and deaths by the time of data collection. For reasons of simplicity, the covariate effects are not displayed in the table. They can be obtained from the analysis documentation at https://osf.io/kws9x/files[43].
*TG* Trust in Government, *EC* Empathic Concern, *FoC* Fear of COVID-19.
** $p < 0.01$; *** $p < 0.001$.

Another question to be addressed concerns the role of generalized trust (i.e., how much people trust each other in general), as much of our argumentation in regard to governmental trust seem also applicable to generalized trust, and since generalized trust has also been found to be a relevant predictor of compliance during the COVID-19 pandemic[63]. Overall, the evidence obtained from our exploratory analysis suggest that the findings obtained with trust in the government are relatively robust against inter-individual and contextual variations of generalized trust, plus specific to the concept of trust in the government. Hence, evidence from the present research emphasizes the unique role of trust in the government and indicates that supporting cooperative action during the pandemic may take more than just citizens' goodwill, and may especially depend on the governments' activities and on how much people find them trustworthy[64]. This is also in line with previous research, confirming the central role that systemic trust has played for cooperative action during the COVID-19 pandemic, by showing that not only interpersonal trust but also – and more strongly – trust in politicians was associated with COVID-19 vaccination[58,65,66].

**Implications**. The present findings underscore that adherence to guidelines aimed at preventing the COVID-19 pandemic spread depends on the interplay between personal individual-level resources (like fear and empathy), and contextual resources (like governmental trust). Mirroring results from extant research[6,21], the present findings highlight the strong and robust role of empathic concern for mobilizing large-scale cooperation that was found to be unaffected by country-level trust in the government. Moreover, the present research underlines that self-centered motives may also play a role for large-scale cooperative endeavors. Fear of disease was associated with individuals' support for COVID-19 containment measures, and that this was especially pronounced when individuals hardly trusted their

governments, or when the general level of governmental trust within a community was low. However, this evidence should not be understood as proof for an underlying causal relationship between governmental distrust and fear-related compliance with COVID-19 mitigating behaviors. Neither should it be regarded as a call for utilizing fear-arousing messages as a way to promote cooperation under such conditions. In fact, while fear of infection may be associated with stronger compliance to COVID-19 health measures[14,20], there is also growing evidence about the negative mental health consequences of enhanced fear of infection[67,68], suggesting that promoting large-scale cooperation via fear would represent a costly and harmful strategy. Promoting empathy, on the contrary, may work as a promising and no-risk strategy to enhance the efficacy of policy recommendations in collective crisis situations across various conditions. However, to benefit from empathy to the utmost, governments should take action in increasing individuals' perceptions of trust in the government.

**Limitations**. The present research allowed us to examine the links between trust and acceptance of imposed restrictions during times of COVID-19, albeit in a correlational rather than causal manner. Though correlational research is central for scientific progress[69], it is but an intermediate step in the process. Correlational research such as the one presented in this study should ideally be supported by ensuing causal confirmations, or at least with corroborated results from independent samples (also see many labs project[70]). Until then, we should interpret the present findings with due caution.

The present data are sufficiently inclusive in terms of cultures of the world, and contain answers from a sufficiently large sample for conducting complex analyses such as hierarchical regression. Nonetheless, there is concern, as is the case usually in survey research[71]. The sampled population might not be representative for the phenomenon and therefore poses questions about the

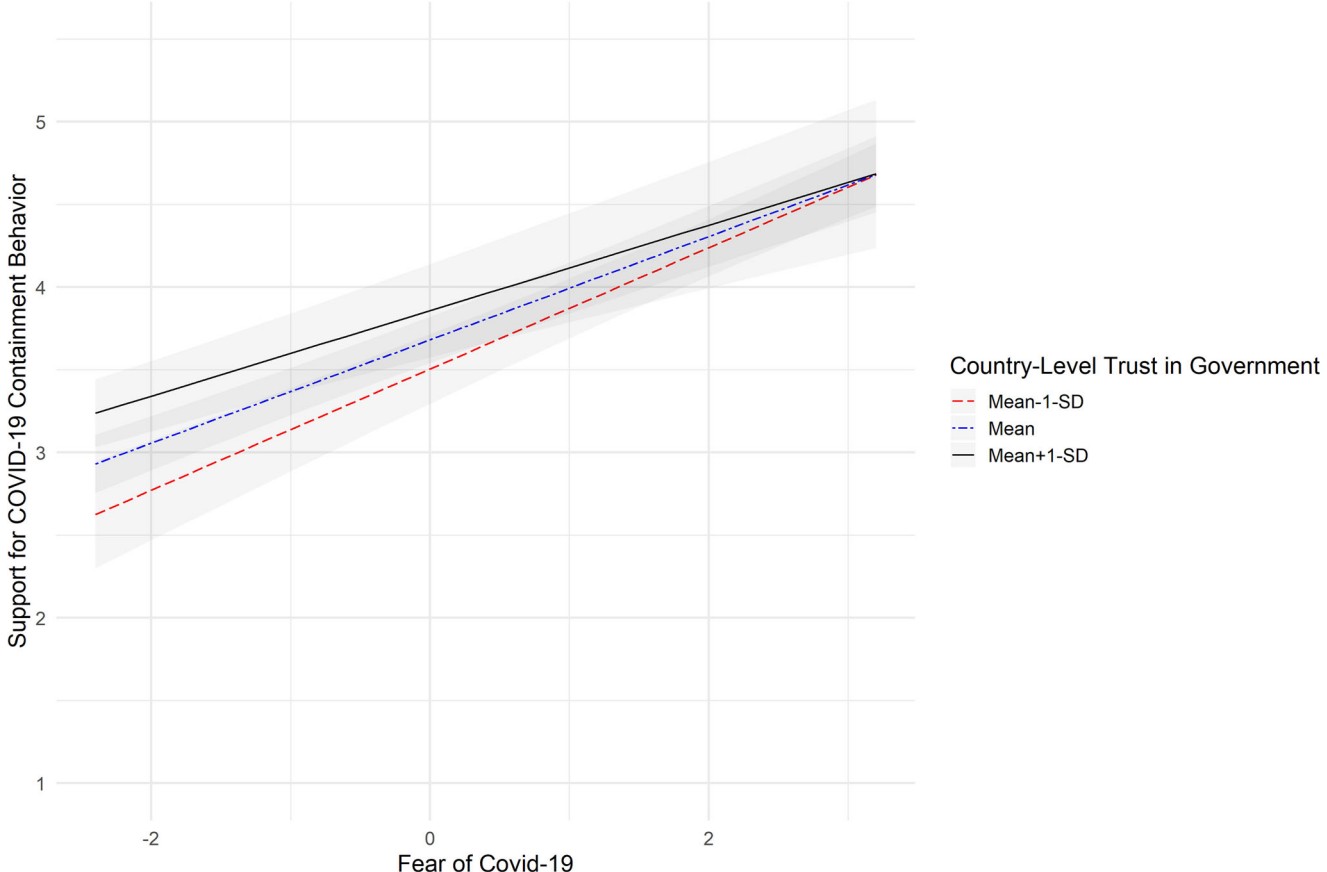

**Fig. 3 Association between fear of COVID-19 and support for COVID-19 containment behaviors under low, medium and high scores of country-level trust in the government.** Notes: The graphic is based on $N = 11,026$ independent responses nested in 34 countries; the black line represents the strength of the association between fear of COVID-19 and support for COVID-19 containment when the country-level trust in the government was high (one standard deviation above the mean); the blue dashed line represents the strength of the association between fear of COVID-19 and support for COVID-19 containment behaviors when the country-level trust in the government was medium; the red dashed line represents the strength of the association between fear of COVID-19 and support for COVID-19 containment behaviors when the country-level trust in the government was low (one standard deviation below the mean). The gray shadowed parts represent the respective 95% Confidence Intervals; support for COVID-19 containment behaviors were assessed on a scale ranging from *1* to *5*, with higher scores representing higher support. The scores for fear of COVID-19 (x-axis) represent the group-mean centered scores.

generalizability of our results. This is especially concerning considering that certain populations who were more at risk, such as older people and people with a migration background, could not be adequately represented. It is noteworthy that the general level of fear of COVID-19 was rather low across the study populations (only in five out of the 34 nations the means for fear of COVID-19 were slightly above the scale midpoint), while the level of empathic concern was consistently high (in all study populations the means for empathy were above the scale midpoint). The results therefore might be biased toward the younger, relatively low-risk and local populations with internet access at the time of data collection.

Moreover, we are aware that the self-reported support for COVID-19 containment behaviors which served as the dependent variable of the present research may not allow conclusions on how individuals would actually behave[72]. For the present research, the choice of a self-report measure was driven by the multi-national nature of the study, and the limited resources available for data collection during the pandemic. Moreover, it should be noted that we deliberately chose to assess the support for practicing COVID-19 containment measures instead of assessing self-reported behavior intentions or past behavior frequencies. That is, because behavior intentions may also be determined by other obligations or situational constraints. For instance, people working in hospitals would not report avoiding hospitals, even though they may consider it important to do so (for others).

Last, we note that the moderation effects, though significant, are small in magnitude, especially the ones obtained from the individual level analysis (e.g., Cohen's $f = 0.02$). However, this is not in itself a concern due to the explorative nature of the study. It could become a concern if these results were taken at face-value without subsequent testing. Hence, even though the findings of the present research seem overall in line with the literature, they may not necessarily apply to other global crises, or if additional independent data were to be collected. We do appreciate the benefits of conducting qualitative research at this stage[73,74], and recommend using both experimental and qualitative research methods to further probe the links between trust and various motivations.

## Data availability
The data generated and/or analyzed during the current study are available on the Open Science Framework repository, https://osf.io/kws9x/[46]. The World Values Survey data are available at https:// www.worldvaluessurvey.org/wvs.jsp[48]. Scores on HDI were extracted

from https://hdr.undp.org/data-center/documentation-and-downloads[55]. Data about hospital beds per thousand were extracted from https://ourworldindata.org/grapher/hospital-beds-per-1000-people[75]. Data about government stringency level were extracted from https://covidtracker.bsg.ox.ac.uk/stringency-map[76]. The scores for daily new COVID-19 cases and deaths represent a 7-days rolling average per million people and were extracted from https://ourworldindata.org/explorers/coronavirus-data-explorer?uniformYAxis=0&pickerSort=asc&pickerMetric=location&Metric=Cases+and+deaths&Interval=7-day+rolling+average&Relative+to+Population=true&Color+by+test+positivity=false[77].

## Code availability

We shared all data and material relevant for this study by using widely-known/standard file formats, and used open tools for data interpretation/re-use. The analysis outputs together with the codes for reproducing these analyses can be accessed under https://osf.io/kws9x/files/[43].

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

## Acknowledgements

We would like to thank Valentina Romo, Jallene Jia En Chua, Melis Yetkin, Emren Burak Ömür, Miguel Landa Blanco, Christine Kraus, Raymond Langley, Youmna Haddad, Abida Sultana, Sadiya Alam Surma, Nishat Tamanna Omi, Asmaul Husna Chara, Monir Hossen, Mohima Akter, Nargees Akter, Khondoker Shahriar Islam, Md Johirul Islam, Joti Saha, Sk.Sabbir Islam Mahi, Tayeba Sultana Sonia, Md. Fakhrul Islam Maruf, Md Mahiuddin Howlader, Shakila Umme Noor, Thaher Uddin Raju, Farha Hossain, Tasfia Tabassum, Md Refat Hossain, Nazia Sultana, Mahbubur Rahman, Meghna Chakravartty, Bikona Ghosh, Airthie Chakma, Esrat Jahan Bani, Khondoker Shahriar Islam, Md Ferdous Shanto, Md. Seikh Sadiul Islam Tanvir, Rowell P. Nitafan, Saima Fariha, Md. Sazzad Hossain, Mahima Ranjan Acharjee, Arnab Bose, Saima Sultana, Istiaq Ahmed, Sami Murshed, Md. Rifat Al Mazid Bhuiyan, Md. Asifur Rahaman, Mst. Fariha Sultana, Kimberly Almonte, Claudia Núñez for helping with the survey translations, data collection and data preparation. We further acknowledge the contribution of the following grants that supported the realization of this research by funding the work of several co-authors. The Cooperative University of Colombia INV3092 provided funding for M.M.-C.; the ANID – Millennium Science Initiative Program (NCS2021_081) provided funding for J.G.; the GESIS-Leibniz Institute for the Social Sciences provided funding for A.S.; the Systemic Risk Institute (LX22NPO5101) funded by European Union - Next Generation EU (Ministry of Education, Youth and Sports, NPO: EXCELES) provided funding for I.P.Š., M.K.-B., and S.S.; the RA Science Committee (Project 21T-5A203) provided funding for H.A., the Council for Scientific and Technological Development (Conselho Nacional de Desenvolvimento Científico e Tecnológico/CNPq - Processo: 401749/2022-3) provided funding for S.G., R.C.F.G., C.M., and J.R. The funders had no role in the design of the study, data collection, preparation of the manuscript or decision to publish.

## Author contributions

A. Karakulak played a lead role in initiating and conceptualizing the project, performing data curation, formal analysis, investigation, methodology design, project administration, visualization, and writing of the original draft, as well as writing review and editing. B.T. played a major role in the conceptualization, investigation, methodology, and writing the original draft and R.D. provided valuable support for the project. M.R. played a supporting role in performing supplemental analyses. These authors contributed equally by supporting the investigation and project administration at the different study sites: M.K.A., P.A., R.A., Y.A.A., A. Amin, D.A.L.A., A. Andres, J.J.B.R.A., M.A., H.A., N.A., M.B.-S., R.K.B., B.B., S.B., D.B., I.B., A.K.B., J.B., K.B., Y.B.-P., C.B., R.C., M.M.C., B.-B.C., G.R.D., D.C.D., A.d.C.D.E., W.G.E., N.F., R.F.-M., J.G., Y.G., S.G., R.C.F.G., M.-T.F., S.G., B.G., J.C.G., M.d.P.G., C.H., G.H., S.H., M.S.H., M.S.I., A.J., N.J., V.J., R.S.K., N.B.A.K., J.K., D.K., Z.K., T.D.K., M.K., R.K., M.K., M.K.-B., A. Kozina, S.E.K., R.L., A.L.-W., Y.-H.L., A.M., S.M., D.M.-M., S.M., B.M., E.A.M., M.M., S.M., J.M.-K., D.M., F.M., R.M.-H., C.M., M.M., P.M., A.N., A.N., F.N., J.N., L.M.A.P., H.O.-A., C.I.O., L.M.O., S.K.M., J.P., I.P., E.A.P., P.P., S.P., F.P., J.R., R.M.R., B.P.D.R., A.S., T.Z.S., D.S., F.S., P.S., S.S., K.S., I.P.Š., O.S.-K., A.S., D.S., L.C.L.S., M.S., J.S., L.F.S., K.S., M.S.S., A.O.S., E.T., L.T.-E., L.D.T., F.U., R.P.V., B.W., G.W.W., P.-J.Y., E.Y., Y.Y., M.A.M.Y. and M.Z.d.S. All authors agreed to the manuscript contents, their authorship and its order.

## Competing interests

J.A. is an Editorial Board Member for Communications Psychology but was not involved in the editorial review of, nor the decision to publish this article. The other authors declare no competing interests.

## Additional information

Arzu Karakulak [1,2 ✉], Beyza Tepe [2], Radosveta Dimitrova[3], Mohamed Abdelrahman [4,5], Plamen Akaliyski[6], Rana Alaseel[7], Yousuf Abdulqader Alkamali[8], Azzam Amin[4], Danny A. Lizarzaburu Aguinaga [9], Andrii Andres [10], John Jamir Benzon R. Aruta[11], Marios Assiotis[12], Hrant Avanesyan [13], Norzihan Ayub [14], Maria Bacikova-Sleskova [15], Raushan Baikanova[16], Batoul Bakkar[7], Sunčica Bartoluci[17], David Benitez[18], Ivanna Bodnar [19], Aidos Bolatov[16], Judyta Borchet[20], Ksenija Bosnar[17], Yunier Broche-Pérez[21], Carmen Buzea [22], Rosalinda Cassibba[23], Marta Martín Carbonell [24], Bin-Bin Chen[25], Gordana Ristevska Dimitrovska[26,27], Dương Công Doanh[28], Alejandra del Carmen Dominguez Espinosa[29], Wassim Gharz Edine[7], Nelli Ferenczi [30], Regina Fernández-Morales [31,32], Jorge Gaete [33,34], Yiqun Gan [35], Suely Giolo [36], Rubia Carla Formighieri Giordani [36], Maria-Therese Friehs [37], Shahar Gindi [38], Biljana Gjoneska [39], Juan Carlos Godoy[40], Maria del Pilar Grazioso[41], Camellia Hancheva[42,43], Given Hapunda[44,45], Shogo Hihara [46,47], Mohd Saiful Husain[48], Md Saiful Islam [49,50], Anna Janovská[17], Nino Javakhishvili[51], Veljko Jovanović [52], Russell Sarwar Kabir [46], Nor Ba'yah Abdul Kadir[53], Johannes Karl[54,55], Darko Katović [17], Zhumaly Kauyzbay [56], Tinka Delakorda Kawashima[46], Maria Kazmierczak[20], Richa Khanna[57], Meetu Khosla[58], Martina Klicperová-Baker [59], Ana Kozina[60], Steven Eric Krauss[61], Rodrigo Landabur[62], Katharina Lefringhausen[63], Aleksandra Lewandowska-Walter[20], Yun-Hsia Liang[64], Ana Makashvili [51], Sadia Malik[65], Denisse Manrique-Millones [66], Stefanos Mastrotheodoros[67,68], Breeda McGrath[69], Enkeleint A. Mechili [70], Marinés Mejía [41], Samson Mhizha [71], Justyna Michalek-Kwiecien [20], Diana Miconi[72], Fatema Mohsen[7,73], Rodrigo Moreta-Herrera [74], Camila Muhl [36], Maria Muradyan [13], Pasquale Musso [23], Andrej Naterer[75], Arash Nemat[76,77], Felix Neto[78], Joana Neto[79], Luz Marina Alonso Palacio[80], Hassan Okati-Aliabad[81], Carlos Iván Orellana [82], Ligia María Orellana[83], Sushanta Kumar Mishra [84], Joonha Park [85], Iuliia Pavlova [19], Eddy Peralta [86], Petro Petrytsa[87], Saša Pišot[88], Franjo Prot[17], José Rasia[36], Rita Rivera[18,89], Benedicta Prihatin Dwi Riyanti[90], Adil Samekin[91], Telman Seisembekov[16], Danielius Serapinas[92], Fabiola Silletti[23], Prerna Sharma[93], Shanu Shukla [94,95], Katarzyna Skrzypińska[20,96], Iva Poláčková Šolcová[59], Olga Solomontos-Kountouri[12], Adrian Stanciu[97], Delia Stefenel[98], Lorena Cecilia López Steinmetz [40,99], Maria Stogianni [100], Jaimee Stuart[101,102], Laura Francisca Sudarnoto[90], Kazumi Sugimura[46], Sadia Sultana[49], Angela Oktavia Suryani[90], Ergyul Tair [103], Lucy Tavitian-Elmadjan[100,104], Luciana Dutra Thome[105], Fitim Uka[106,107], Rasa Pilkauskaitė Valickienė[92], Brett Walter[46], Guilherme W. Wendt [108], Pei-Jung Yang[109], Ebrar Yıldırım [110], Yue Yu[111,112], Maria Angela Mattar Yunes[113], Milene Zanoni da Silva [114] & Maksim Rudnev[115]

[1]Istanbul Policy Center, Sabanci University, Istanbul, Turkey. [2]Department of Psychology, MEF University, Istanbul, Turkey. [3]Department of Psychology, Stockholm University, Stockholm, Sweden. [4]Social Psychology Department, Doha Institute for Graduate Studies, Doha, Qatar. [5]Mokhtass for Consultations and Research, Doha, Qatar. [6]Department of Sociology and Social Policy, Lingnan University, Hong Kong SAR, China. [7]Faculty of Medicine, Syrian Private University, Damascus, Syrian Arab Republic. [8]United Private School, Athaiba, Oman. [9]Universidad César

Vallejo, Trujillo, Peru. [10]Department of Physical Education, Lviv Polytechnic National University, Lviv, Ukraine. [11]Department of Psychology, De La Salle University, Manila, Philippines. [12]Theological School, Church of Cyprus, Nicosia, Cyprus. [13]General Psychology Chair, Yerevan State University, Yerevan, Armenia. [14]Faculty of Psychology and Education, University of Malaysia Sabah, Sabah, Malaysia. [15]Department of Educational Psychology and Health Psychology, Pavol Jozef Šafárik University, Košice, Slovakia. [16]Astana Medical University, Astana, Kazakhstan. [17]University of Zagreb, Zagreb, Croatia. [18]Clinical Psychology Department, Albizu University, Miami, FL, USA. [19]Lviv State University of Physical Culture, Lviv, Ukraine. [20]Institute of Psychology, University of Gdansk, Gdansk, Poland. [21]Department of Psychology, Universidad Central Marta Abreu de Las Villas, Santa Clara, Cuba. [22]Department of Social Sciences and Communication, Transilvania University of Brasov, Brasov, Romania. [23]Department of Education, Psychology, Communication, University of Bari, Bari, Italy. [24]Faculty of Psychology, Cooperative University of Colombia, Santa Marta, Colombia. [25]Psychology Department, Fudan University, Shanghai, China. [26]Higher Medical School, University St. Kliment Ohridski, Bitola, North Macedonia. [27]PHI Psihomedika, Bitola, North Macedonia. [28]National Economics University, Hanoi, Vietnam. [29]Iberoamerican University, Mexico City, Mexico. [30]Department of Life Sciences, College of Health, Medicine and Life Sciences, Brunel University, London, UK. [31]Psychology Department, Universidad Francisco Marroquín, Guatemala City, Guatemala. [32]Humanities Department, Universidad Rafael Landívar, Guatemala City, Guatemala. [33]Faculty of Education, Universidad de los Andes, Santiago, Chile. [34]Millennium Nucleus to Improve the Mental Health of Adolescents and Youths, Imhay, Santiago, Chile. [35]School of Psychological and Cognitive Sciences and Beijing Key Laboratory of Behavior and Mental Health, Peking University, Beijing, China. [36]Federal University of Parana, Curitiba, Brazil. [37]Faculty for Psychology, Fern Universität, Hagen, Germany. [38]Faculty of Education, Beit Berl College, Kfar Sava, Israel. [39]Macedonian Academy of Sciences and Arts, Skopje, North Macedonia. [40]Psychological Research Institute (IIPsi), National University of Córdoba - CONICET, Córdoba, Argentina. [41]Universidad del Valle de Guatemala, Proyecto Aigle Guatemala, Cdad. de Guatemala, Guatemala. [42]Department of General, Experimental, Developmental, and Health Psychology, Sofia University "St. Kliment Ohridski", Sofia, Bulgaria. [43]Center for Psychological Counselling and Research, Sofia University "St. Kliment Ohridski", Sofia, Bulgaria. [44]Psychology Department, University of Zambia, Lusaka, Zambia. [45]Impact Managers, Lusaka, Zambia. [46]Hiroshima University, Hiroshima, Japan. [47]Faculty of Business Administration, Matsuyama University, Ehime, Japan. [48]University Sains Malaysia, Penang, Malaysia. [49]Department of Public Health and Informatics, Jahangirnagar University, Dhaka, Bangladesh. [50]Centre for Advanced Research Excellence in Public Health, Dhaka, Bangladesh. [51]Dimitri Uznadze Institute of Psychology, Ilia State University, Tbilisi, Georgia. [52]Department of Psychology, University of Novi Sad, Novi Sad, Serbia. [53]Center for Research in Psychology and Human Well-being, Universiti Kebangsaan Malaysia, Bangi Selangor, Malaysia. [54]School of Psychology, Victoria University of Wellington, Wellington, New Zealand. [55]School of Psychology, Dublin City University, Dublin, Ireland. [56]South Kazakhstan Medical Academy, Shymkent, Kazakhstan. [57]School of Human Ecology, Tata Institute of Social Sciences, Mumbai, Maharashtra, India. [58]University of Delhi, Delhi, India. [59]Institute of Psychology, Czech Academy of Sciences, Prague, Czech Republic. [60]Center for Evaluation Studies, Educational Research Institute, Ljubljana, Slovenia. [61]Institute for Social Science Studies, Universiti Putra Malaysia, Serdang-Selangor, Malaysia. [62]Departamento de Psicología, Universidad de Atacama, Copiapó, Chile. [63]Department of Psychology, Heriot-Watt University, Edinburgh, UK. [64]Department of Education, University of Taipei, Taipei, Taiwan. [65]Department of Psychology, University of Sargodha, Sargodha, Pakistan. [66]Carrera de Psicología, Universidad Científica del Sur, Lima, Peru. [67]Department of Youth and Family, Utrecht University, Utrecht, The Netherlands. [68]Department of Psychology, University of Crete, Rethymno, Greece. [69]The Chicago School of Professional Psychology, Chicago, IL, USA. [70]Department of Healthcare, Faculty of Health, University of Vlora, Vlore, Albania. [71]Department of Applied Psychology, University of Zimbabwe, Harare, Zimbabwe. [72]Department of Educational Psychology and Andragogy, University of Montreal, Montreal, QC, Canada. [73]Ipswich Hospital, Ipswich, UK. [74]School of Psychology, Pontifical Catholic University of Ecuador, Ambato, Ecuador. [75]University of Maribor, Maribor, Slovenia. [76]Kabul University of Medical Sciences, Kabul, Afghanistan. [77]Karolinska Institutet, Solna, Sweden. [78]Department of Psychology, University of Porto, Porto, Portugal. [79]REMIT - Research on Economics, Management and Information Technologies, Universidade Portucalense, Porto, Portugal. [80]Universidad del Norte, Division of Health Sciences, Barranquilla, Colombia. [81]Health Promotion Research Center, Zahedan University of Medical Sciences, Zahedan, Iran. [82]Social Sciences Doctoral and Master Program, Don Bosco University, Antiguo Cuscatlan, El Salvador. [83]Núcleo Científico-Tecnológico en Ciencias Sociales y Humanidades, Universidad de La Frontera, Temuco, Chile. [84]Indian Institute of Management Bangalore, Bangalore, India. [85]School of Management, NUCB Business School, Nagoya, Japan. [86]Medicine School, Mother and Teacher Pontifical Catholic University, Santiago, Dominican Republic. [87]Department of Physical Education and Rehabilitation, Ternopil Volodymyr Hnatiuk National Pedagogical University, Ternopil, Ukraine. [88]Institute for Kinesiology Research, Science and Research Centre Koper, Koper, Slovenia. [89]Counseling & Psychological Services, Duke University, Durham, NC, USA. [90]Faculty of Psychology, Atma Jaya Catholic University of Indonesia, Jakarta, Indonesia. [91]School of Liberal Arts, M. Narikbayev KAZGUU University, Astana, Kazakhstan. [92]Institute of Psychology, Mykolas Romeris University, Vilnius, Lithuania. [93]Clinical Psychology Department, Institute of Human Behavior and Allied Sciences, Delhi, India. [94]Indian Institute of Management Indore, Indore, India. [95]Interdisciplinary Research Team on Internet and Society, Faculty of Social Studies, Masaryk University, Brno, Czech Republic. [96]Polish Society for the Psychology of Religion, University of Gdańsk, Gdańsk, Poland. [97]Data and Research on Society, GESIS-Leibniz Institute for the Social Sciences, Mannheim, Germany. [98]Faculty of Social Science and Humanities, Lucian Blaga University of Sibiu, Sibiu, Romania. [99]Siglo 21 University, Córdoba, Argentina. [100]Department of Culture Studies, Tilburg University, Tilburg, The Netherlands. [101]United Nations University, Macau, Macau SAR. [102]School of Applied Psychology, Griffith University, Brisbane, QLD, Australia. [103]Institute for Population and Human Studies, Bulgarian Academy of Sciences, Sofia, Bulgaria. [104]Faculty of Social and Behavioral Sciences, Department of Psychology, Haigazian University, Beirut, Lebanon. [105]Federal University of Bahia, Salvador, Brazil. [106]University of Prishtina "Hasan Prishtina", Prishtina, Kosovo. [107]Department of Psychology, Multidisciplinary Clinic Empatia, Prishtina, Kosovo. [108]Western Paraná State University, Francisco Beltrão, Brazil. [109]Graduate Institute of Social Work, National Chengchi University, Taipei, Taiwan. [110]Yeditepe University, Istanbul, Turkey. [111]Centre for Research in Child Development, National Institute of Education, Nanyang Technological University, Singapore, Singapore. [112]Singapore Centre for Character and Citizenship Education, National Institute of Education, Nanyang Technological University, Singapore, Singapore. [113]Universidade Salgado de Oliveira, Niterói, Brazil. [114]Universidade Estadual de Ponta Grossa (UEPG), Ponta Grossa, Parana, Brazil. [115]Department of Psychology, University of Waterloo, Waterloo, ON, Canada. ✉email: arzuaydinlikarakulak@gmail.com; karakulaka@mef.edu.tr

