## [Peer Review File · Communications Psychology]

15th Mar 23

Dear Dr Karakulak,

Thank you for your patience during the peer-review process. Your manuscript titled "Empathy, Fear of Disease and Support for COVID-19 Containment Behaviors: Evidence from 34 Countries on the Moderating Role of Governmental Trust" has now been seen by 3 reviewers, and I include their comments at the end of this message. They find your work of interest, but raised some important points. We are interested in the possibility of publishing your study in *Communications Psychology*, but would like to consider your responses to these concerns and assess a revised manuscript before we make a final decision on publication.

We therefore invite you to revise and resubmit your manuscript, along with a point-by-point response to the reviewers. Please highlight all changes in the manuscript text file.

Editorially, we ask you to prioritize two issues in revision:

1) All reviewers raise concern regarding the correlational nature of the data, a concern you also note in the Limitations section. To appropriately revise the manuscript, we ask you to remove any directional language, highlighting the correlational character throughout all sections (Abstract, Introduction, Discussion). You may include the proposed mechanism that led to your hypothesis, but you must allow for the fact that alternative mechanisms would be equally well supported by the data (see Reviewer #2).

2) To strengthen the work, we further ask you to perform the control analyses requested by Reviewer #3 and we recommend preregistration of this analysis. Please note our general guidelines regarding preregistration, <https://www.nature.com/commspsychol/submit/preregistration>. Please highlight that your hypotheses, but not the exact implemented analysis were preregistered.

Please use the following link to submit your revised manuscript, point-by-point response to the referees' comments (which should be in a separate document to any cover letter) and the completed checklist:

[Link redacted]

We understand that due the situation caused by the recent Turkey-Syria earthquake, the time required for revision may be longer than usual. We would appreciate it if you could keep us informed about an estimated timescale for resubmission, to facilitate our planning. Of course, if you are unable to estimate, we are happy to accommodate necessary extensions nevertheless.

Please do not hesitate to contact me if you have any questions or would like to discuss these revisions further. We look forward to seeing the revised manuscript and thank you for the opportunity to review your work.

Best regards,

Antonia Eisenkoeck

Antonia Eisenkoeck
Senior Editor
Communications Psychology

EDITORIAL POLICIES AND FORMATTING

Editorial Policy: [Policy requirements](https://www.nature.com/documents/nr-editorial-policy-checklist.pdf) (Download the link to your computer as a PDF.)

Furthermore, please align your manuscript with our format requirements, which are summarized on the following checklist:

[Communications Psychology formatting checklist](https://www.nature.com/documents/commsj-psychol-style-formatting-checklist-article.pdf)

and also in our style and formatting guide [Communications Psychology formatting guide](https://www.nature.com/documents/commsj-psychol-style-formatting-guide-accept.pdf) .

* **CODE AVAILABILITY:** All Communications Psychology manuscripts must include a section titled "Code Availability" at the end of the methods section. In the event of publication, we require that the custom analysis code supporting your conclusions is made available in a publicly accessible repository; at publication, we ask you to choose a repository that provides a DOI for the code; the link to the repository and the DOI will need to be included in the Code Availability statement. Publication as Supplementary Information will not suffice. We ask you to prepare code at this stage, to avoid delays later on in the process.

* **DATA AVAILABILITY:**

All Communications Psychology manuscripts must include a section titled "Data Availability" at the end of the Methods section or main text (if no Methods). More information on this policy, is available at [http://www.nature.com/authors/policies/data/data-availability-statements-data-](http://www.nature.com/authors/policies/data/data-availability-statements-data-citations.pdf)

citations.pdf

At a minimum the Data availability statement must explain how the data can be obtained and whether there are any restrictions on data sharing. Communications Psychology strongly endorses open sharing of data. If you do make your data openly available, please include in the statement:

We recommend submitting the data to discipline-specific, community-recognized repositories, where possible and a list of recommended repositories is provided at <http://www.nature.com/sdata/policies/repositories>.

If a community resource is unavailable, data can be submitted to generalist repositories such as [figshare](https://figshare.com/) or [Dryad Digital Repository](http://datadryad.org/). Please provide a unique identifier for the data (for example a DOI or a permanent URL) in the data availability statement, if possible. If the repository does not provide identifiers, we encourage authors to supply the search terms that will return the data. For data that have been obtained from publicly available sources, please provide a URL and the specific data product name in the data availability statement. Data with a DOI should be further cited in the methods reference section.

REVIEWERS' EXPERTISE:

Reviewer #1: compliance with COVID-19 measures
Reviewer #2: compliance with COVID-19 measures
Reviewer #3: World Value Study, statistics

REVIEWERS' COMMENTS:

Reviewer #1 (Remarks to the Author):

The manuscript is very well written and it presents a unique data set. It is not only reporting a multinational study, but also on that involves a wider variety of countries than most similar datasets. Here, most of the data does clearly not stem from Western industrialized countries, which is clearly an asset.

Another strength is that the hypotheses have been preregisters. Unfortunately, the quality of the preregistration is lower regarding the data analysis procedure.

Testing the moderation of the relations between COVID-19 containment behaviors and fear as well as empathy by governmental trust is to the best of my knowledge innovative. At the same time, the number of studies testing predictors of COVID-19 containment behaviors is almost countless and the main effects of empathy of fear, which have long been known, are much larger than the interactions with trust.

In my opinion, standard effect sizes (e.g., Cohen's d) should be added for all reported effects. This

would help to understand whether the $B = .002$ for the TG x FoC interaction is meaningful in size. My sense is, that it might be not. Also, tables could profit from some additional information (e.g., Table 5 does not indicate, which type of indices are reported). Finally, numbers should be reported with 2 meaningful digits (e.g., in table 5 some SE = .00 which is not informative).

As acknowledged by the authors the reported data is cross-sectional, which limits the conclusions that can be drawn about causality. A longitudinal design would be hard to implement in a multinational study. Still it is hard to draw implications from cross-sectional data.

Overall, I like the idea, the way the paper is written, the data, and the findings and I strongly believe it should be published in *Communications Psychology* or somewhere else.

Reviewer #2 (Remarks to the Author):

Reviewer report: Empathy, Fear of Disease and Support for COVID-19 Containment Behaviors: Evidence from 34 Countries on the Moderating Role of Governmental Trust

The paper investigates the association between empathy and fear on one hand and support for Covid containment measures on the other in a large data collection exercise spanning 34 countries. The article moreover investigates and uncovers the moderating effect of governmental trust on the relationship under investigation.

The findings report a strong and positive correlation between support for containment measures and fear and empathy with a strong moderating role of trust in government, whereby strong trust in government is associated with a strong empathy-support correlation and low trust in government is associated with a strong fear-support correlation. I believe this evidence, though I do not agree with the mechanism the author propose as explanation for it.

I struggle with the logic behind the proposed relationship between trust in government and empathy-driven cooperation (page 13). This logic is key in the paper as the interpretation of the results the authors obtain rests entirely on it. The authors argue that "Only when trust in the government is high, individuals would perceive their individual efforts to contain the virus as positive for the community at large, because preventing the spread of the COVID-19 pandemic would be perceived as a concerted effort and a shared goal by many. On the other hand, when there is no trust or little trust in the government, individuals would not expect that their individual compliance with public health measures will produce any public, other-oriented benefit, as they would not expect their contribution to be reciprocated by their government's actions and regulations."

If this logic holds, then the interpretation of the slopes in the interacted regression presented holds (with the caveat that the causal relationship remains unobservable and can only be taken with a pinch of salt and only to the extent with which we believe in the logic presented above).

However, as I hinted at, the logic is not fully convincing. If regulations are in place, why would I not comply in fear of the government not implementing regulations? (this question paraphrases the authors' words, if I understand them correctly)

Here's an alternative mechanism which to my understanding would produce the same observations reported in this study.

Trust in government and generalised social trust are suspected to be (to the least) correlates (see for instance Putnam or Sønderskov et al. 2016). If I state I trust the government, it is likely I trust others in society too. If I am empathic, I will support covid 19 measures because my belief is that others in society will comply with those (enacted) regulations (Hypothesis 1a is confirmed). If I don't trust the government and hence neither I trust others, the only reason I would want more stringent regulation is my fear that people won't comply (because I don't trust them) and hence stricter rules

are required to get a minimum amount of compliance (Hypothesis 2a is confirmed). The same logic would produce arguments close to the above and would result in statements which would confirm Hypotheses 1b and 2b as well.

The authors are open about the correlational nature of their study, but only on page 24, and the entire paper reads as if trust in government is indeed the moderator between empathy/fear and support for governmental policies. That the study is correlational is not my concern here. What I am concerned with is that acceptance of their theory rests upon assumptions (the logic reported above) which in my view are not very convincing.

Should the authors have individual data on social trust they could replicate their analyses to check whether models including social trust are more explanatory than those including governmental trust. Perhaps some statistical technique might be able to (perhaps imperfectly) disentangle the two (I fear no suggestion comes to my mind). Country level data on social trust is publicly available. Perhaps they could include analogous analyses to the mixed-level ones using this data.

Reviewer #3 (Remarks to the Author):

Referee Report - Empathy, Fear of Disease and Support for COVID-19 Containment Behaviors: Evidence from 34 Countries on the Moderating Role of Governmental Trust

Summary of the findings

Using a large sample of 12,758 individuals from 34 countries, the authors show that empathic prosocial concern and fear of disease are positively and significantly related with support towards preventive COVID-19 behaviors. Further, the authors emphasize that these relationships are moderated by the level of trust in the countries' government. Finally, the authors discuss how both fear and empathy motivations to support preventive COVID-19 behaviors may be shaped by the socio-cultural context.

Comments

I appreciated the opportunity to read the paper. Overall, the paper is well-written and presents some interesting findings. Yet, I have a couple of points (especially related to methodology) that I would like to elaborate on in more detail.

1. I like how the authors carry out their main analysis; however, I wonder why the authors do not control for the individual characteristics of the respondents (e.g., age and gender). What is the reason for that?
2. Similarly, I would suggest to the authors to include a control variable for the severeness of COVID-19 in the respective countries? For example, the number of cases/death per inhabitants.
3. Further, I would suggest to the authors to use a different regression specification for robustness purposes. Specifically, I would suggest using a specification including country fixed effects and clustering the standard errors across countries. This would help to address endogeneity concerns.
4. I like that the authors use the country-level scores for trust in the government from the World Values Survey. However, Engelhardt et al. (2021) also investigated a related question in the finance

literature and used scores from the OECD for robustness checks. I would suggest to the authors to check robustness by using these scores as well.

5. I also recommend to the authors to check if there is a moderating effect when focusing the level of societal trust. In this respect, Engelhardt et al. (2021) note that “trust in governments might only be one side of the coin as trust in fellow citizens obeying the government’s orders might also be of significant importance.” The data on societal trust is also available from the World Values Survey.

RESPONSE LETTER

Reviewer 1 Comments:

General comment:

The manuscript is very well written and it presents a unique data set. It is not only reporting a multinational study, but also on that involves a wider variety of countries than most similar datasets. Here, most of the data does clearly not stem from Western industrialized countries, which is clearly an asset. Another strength is that the hypotheses have been preregisters. Unfortunately, the quality of the preregistration is lower regarding the data analysis procedure. Testing the moderation of the relations between COVID-19 containment behaviors and fear as well as empathy by governmental trust is to the best of my knowledge innovative. At the same time, the number of studies testing predictors of COVID-19 containment behaviors is almost countless and the main effects of empathy of fear, which have long been known, are much larger than the interactions with trust.

Response: We are happy that the Reviewer find merits in our work, and grateful for their compliments on the clarity of our presentation, our mindfulness to include data from less-represented countries, the innovative analytical approaches, and our open-science practices (with pre-registration of our hypotheses). We are using this opportunity to assure that we took care to address all remaining issues. In line with the reviewer's suggestion, we updated our preregistration by adding a detailed analysis plan (see <https://osf.io/k2wjr>). For the remainder of the issues, please refer to our detailed responses below.

1. In my opinion, standard effect sizes (e.g., Cohen's d) should be added for all reported effects. This would help to understand whether the $B = .002$ for the TG x FoC interaction is meaningful in size. My sense is, that it might be not.

Response: We agree with the Reviewer and we added effect sizes (Cohen's f) for all estimates, when reporting the main effects and interactions in the main text (see highlighted parts in the results section on page 19-22). As already noted by the Reviewer, the effect sizes for the interaction were relatively small (especially for the individual level analyses). Therefore, we drew additional attention to this limitation in the discussion of the results on page 23/24 and page 28/29.

Page 23/24:

It should be noted however, that even though trust in one's government altered the strength of the association between empathic concern and support for COVID-19 containment behaviors, the strength of the association under high versus low trust in the government was almost identical (Cohen's $f = 2.01$ vs. Cohen's $f = 1.99$). Thus, even though the association between empathy and support for large-scale cooperative action may become

more pronounced when individuals perceive their governments as more (compared to less) trustworthy, such interplay with trust seems less salient for empathic concern (compared to fear of COVID-19).

Page 28/29:

Last, we note that the moderation effects, though significant, are small in magnitude, especially the ones obtained from the individual level analysis (Cohen's $f = 0.02$).

However, this is not in itself a concern due to the explorative nature of the study. It could become a concern if these results were taken at face-value without subsequent testing

2. Also, tables could profit from some additional information (e.g., Table 5 does not indicate, which type of indices are reported). Finally, numbers should be reported with 2 meaningful digits (e.g., in table 5 some SE = .00 which is not informative).

Response: Following this recommendation we updated the tables to include: a) Legend with information regarding the reported indices; b) Brief description as regards the performed analyses and; c) Correct reporting of numbers (with 2 meaningful digits).

3. As acknowledged by the authors the reported data is cross-sectional, which limits the conclusions that can be drawn about causality. A longitudinal design would be hard to implement in a multinational study. Still it is hard to draw implications from cross-sectional data. Overall, I like the idea, the way the paper is written, the data, and the findings and I strongly believe it should be published in Communications Psychology or somewhere else.

We are very grateful for this encouraging feedback, the constructive approach with overall positive evaluation of our work, and the recommendation for publication of our results. In the revised version of the manuscript, we paid more attention to avoid language that would imply causality. Furthermore, we would like to reiterate, that the cross-sectional approach was highlighted as a possible limitation at the very beginning of the corresponding "Limitation" section of the manuscript (on page 27), so as to acknowledge that caution is warranted when interpreting the results and to caution the audience against making overconfident conclusions. The statement reads as follows:

Page 27:

The present research allowed us to examine the links between trust and acceptance of imposed restrictions during times of COVID-19, albeit in a correlational rather than causal manner. Though correlational research is central for scientific progress⁵⁸, it is but an intermediate step in the process. Correlational research such as the one presented in this study should ideally be supported by ensuing causal confirmations, or at least with corroborated results from independent samples (also see many labs project⁵⁹). This of course is not a feasible solution in the present situation, as, hopefully, the COVID-19

pandemic is nearing its end. It is, however, an observation that should make us interpret the present findings with due caution.

General comment:

The findings report a strong and positive correlation between support for containment measures and fear and empathy with a strong moderating role of trust in government, whereby strong trust in government is associated with a strong empathy-support correlation and low trust in government is associated with a strong fear-support correlation. I believe this evidence, though I do not agree with the mechanism the author propose as explanation for it.

Response: We thank the Reviewer for the concise summary of our work, and correct interpretation of our findings. Also, we acknowledge the issues raised in the general comment and we made sure to address them in our responses below.

I struggle with the logic behind the proposed relationship between trust in government and empathy-driven cooperation (page 13). This logic is key in the paper as the interpretation of the results the authors obtain rests entirely on it. The authors argue that “Only when trust in the government is high, individuals would perceive their individual efforts to contain the virus as positive for the community at large, because preventing the spread of the COVID-19 pandemic would be perceived as a concerted effort and a shared goal by many. On the other hand, when there is no trust or little trust in the government, individuals would not expect that their individual compliance with public health measures will produce any public, other-oriented benefit, as they would not expect their contribution to be reciprocated by their government’s actions and regulations.” If this logic holds, then the interpretation of the slopes in the interacted regression presented holds (with the caveat that the causal relationship remains unobservable and can only be taken with a pinch of salt and only to the extent with which we believe in the logic presented above).

Response: We thank the Reviewer for their observation that the rationale for our hypotheses seemed less convincing in the first version of the manuscript. We believe that this was partly related to the fact that some parts of the justification did not sufficiently outline the unique role of trust in the government (as opposed to generalized trust). We extensively revised both the introduction and the discussion of the paper to reduce such confusion, and to strengthen the rationale for our hypotheses. Additionally, we ran supplemental analyses to address concerns raised by the Reviewer. We will outline these changes in detail below, and hope that the current version of the manuscript addresses the reviewer’s concerns in a manner that is both clear, concise and comprehensive.

1. However, as I hinted at, the logic is not fully convincing. If regulations are in place, why would I not comply in fear of the government not implementing

regulations? (this question paraphrases the authors' words, if I understand them correctly)

Response: We totally agree with the issue raised by the Reviewer here. Actually, we did not hypothesize that fear would not operate as a motivator if regulations are in place. However, we agree that the way the argument was developed (in the original version) opened space for such an understanding. To clarify that our hypotheses would not imply that fear will become irrelevant (or less relevant than empathy) when trust is high, we added the following clarification to our manuscript on page 25 (discussion):

Page 25:

Yet, it should be noted that our results do not indicate that empathy would not operate in low trust settings, or that fear would not operate in high trust settings. Instead, our results support the view that both motivational pathways seem relevant for mobilizing cooperative action across individuals and contexts (i.e., as main effects were consistently recorded across various levels of governmental trust), but suggest that empathy-driven cooperation is likely to unfold optimally when trust in the government is perceived as high, while fear-driven cooperation is likely to become most pronounced in contexts where trust in the government is generally low, or when trust in the government is individually perceived as low.

2. Here's an alternative mechanism which to my understanding would produce the same observations reported in this study. Trust in government and generalised social trust are suspected to be (to the least) correlates (see for instance Putnam or Sønderskov et al. 2016). If I state I trust the government, it is likely I trust others in society too. If I am empathic, I will support covid 19 measures because my belief is that others in society will comply with those (enacted) regulations (Hypothesis 1a is confirmed). If I don't trust the government and hence neither I trust others, the only reason I would want more stringent regulation is my fear that people won't comply (because I don't trust them) and hence stricter rules are required to get a minimum amount of compliance (Hypothesis 2a is confirmed). The same logic would produce arguments close to the above and would result in statements which would confirm Hypotheses 1b and 2b as well.

Response: We thank the Reviewer for this insightful observation, and for employing additional efforts to provide alternative interpretation of (and explanation for) our findings. And we agree that in the previous version of the manuscript the unique role of trust in the government (compared to trust in general) was not sufficiently outlined. We addressed this issue by revising the introduction extensively (p. 14-16) and hope that these revisions address the concerns raised by the reviewer.

Page 14-16:

As outlined above, mobilizing people for large-scale and long-term cooperative acts (as in times of the pandemic) is challenging; mainly due to the high level of uncertainty it involves. Some individuals (especially those who feel less vulnerable to the virus) might even experience a social dilemma whereby cooperating would require them to sacrifice their concrete, short-term, and selfish interests (e.g., by avoiding social events, by face masking, etc.) over the more abstract, less traceable, and long-term goal of protecting other (mostly vulnerable) people from infection. We argue that such a dilemma should become especially salient, and thus hinder empathy-driven cooperation, under highly unpredictable conditions where one is doubtful about whether or not individual “sacrifices” will promote any long-term public benefit. The way in which people perceive their governments, namely whether they view them as competent, protective, and caring – simply, whether they trust in their government or not – may represent one of the factors that guides people’s perceptions of (un)predictability. Namely, when people anticipate their governments to be supportive and to take the needed measures for protecting their citizens from any threat (including the COVID-19 pandemic), individuals would become more confident that their selfless and empathy-driven activities will truly result in positive public health outcomes.

On the contrary, when there is no trust or little trust in the government, unpredictability will increase, and individuals will doubt whether their individual compliance with public health measures will produce any public, other-oriented benefit. Even if respective measures are implemented by the government, those with little trust in the government would be propelled to suspect that they may be implemented in a superficial and non-transparent manner. For instance, the value of face masking (individually and even collectively) to protect others from infection would become less obvious and more questionable when governments do not offer the needed framing conditions and (are perceived to) act in pandemic-ignorant ways.

However, under such volatile conditions cooperation may still occur, yet for different sets of reasons. As outlined above, the compliance with safe and preventive COVID-19 behaviors is not only an act that may be performed for the sake of protecting others from infection, but also has direct (beneficial) implications on personal health, and may thus be considered as a self-protective act. In circumstances where trust in the government is low, people would experience higher uncertainty, feel more vulnerable and susceptible to the virus, and their cooperation would be more strongly driven by selfish concerns such as fear of COVID-19. Therefore, we propose that the relationship between empathic concern and support for COVID-19 containment behaviors will become stronger when the government is (generally) perceived as more (compared to less) trustworthy, and that conversely the association between fear of disease and support for COVID-19 containment behaviors will become stronger when the government is (generally) perceived as less (compared to more) trustworthy.

1. The authors are open about the correlational nature of their study, but only on page 24, and the entire paper reads as if trust in government is indeed the moderator between empathy/fear and support for governmental policies. That the

study is correlational is not my concern here. What I am concerned with is that acceptance of their theory rests upon assumptions (the logic reported above) which in my view are not very convincing. Should the authors have individual data on social trust they could replicate their analyses to check whether models including social trust are more explanatory than those including governmental trust. Perhaps some statistical technique might be able to (perhaps imperfectly) disentangle the two (I fear no suggestion comes to my mind). Country level data on social trust is publicly available. Perhaps they could include analogous analyses to the mixed-level ones using this data.

Response: We thank the Reviewer for this excellent suggestion, which actually helped us to strengthen our argument. Accordingly, we conducted additional analyses with generalized trust (both at the individual and country level) and discussed their results in the discussion on p 25/26 The excerpt is copied as follows:

Page 25/26:

Another question to be addressed concerns the role of generalized trust (i.e., how much people trust each other in general), as much of our argumentation in regard to governmental trust seem also applicable to generalized trust, and since generalized trust has also been found to be a relevant predictor of compliance during the COVID-19 pandemic⁵². Yet —while we acknowledge the crucial role of generalized trust— the present research deemed important to focus on trust in the government, since pandemic management obviously takes more than just citizens' goodwill, and largely depends on the governments' activities and on how much people find them trustworthy⁵³. This is also in line with previous research, confirming the central and unique role that “systemic trust” has played for cooperative action during the COVID-19 pandemic, by showing that not only interpersonal trust ($r = 0.14, p < 0.05$) but also – and more strongly – trust in politicians was associated with COVID-19 vaccination ($r = 0.24, p < 0.01$)^{47,54,55}. Nevertheless, we conducted additional analyses to examine (1) whether our findings remain robust when generalized trust is added as another covariate to the analyses, and (2) whether the proposed moderation effects can also be obtained with generalized (instead of governmental) trust, using the same analytical procedures as described above. The scores for individual-level generalized trust were obtained by averaging the scores that respondents provided to the following two questions: “In general, most people in our community can be trusted” and “Most people in our community are fair and do not take advantage of you” ($I =$ totally disagree, $5 =$ totally agree, $r = 0.78$). The scores for country-level generalized trust, were extracted from the most recent wave of the World Values Survey (WVS).

First, our analyses revealed that the results did not change when generalized trust was added as a covariate into the regression models. Second, when analyses were conducted with generalized instead of governmental trust, no interaction with fear and empathy emerged, neither on the individual-level nor the country-level analyses (expect for a marginal significant interaction with fear of COVID-19 in the proposed direction

with $p = 0.089$). Interestingly, generalized trust was also not associated directly with support for COVID-19 containment measures, neither on an individual level ($p = 0.073$) nor on a country level ($p = 0.204$). This supports the view that trust in the government seems conceptually distinct from generalized trust (in our research they were correlated with $r = 0.58$ on the individual-level; and $r = 0.10$ on at the country-level), exhibiting unique associations with support for COVID-19 containment behaviors (detailed results are available at <https://osf.io/kws9x/files/> under supplementary analyses).

Reviewer 3 Comments:

General comment

I appreciated the opportunity to read the paper. Overall, the paper is well-written and presents some interesting findings. Yet, I have a couple of points (especially related to methodology) that I would like to elaborate on in more detail.

Response: Thank you, we truly appreciate your kind words and very concrete instructions that helped us improve the presentation of our findings.

1. I like how the authors carry out their main analysis; however, I wonder why the authors do not control for the individual characteristics of the respondents (e.g., age and gender). What is the reason for that?

Response: Thank you for drawing attention to this issue. The reason for this choice was simply due to parsimony. In the original version of the paper, these analyses were actually conducted and reported in a footnote (the output was made available via the OSF). As the results did not change substantially, we decided to present results without covariate effects. However, in the revised version of the paper, we present the analyses and results by controlling for the effects of gender, age, HDI, hospital beds per 1000, data collection month, government stringency level, COVID-19 cases and death rates by the time of data collection (as kindly suggested by you and the Editor). In short, our findings remain even after controlling for the above mentioned socio-demographic variables. This is noted inside the manuscript (on page 19 and in Table 4 & 5) as follows:

Page 19:

Additionally, to account for possible socio-demographic, country- and pandemic-specific effects, we entered respondents' age and their gender, and country-level scores of the Human Development Index (HDI, a composite score reflecting the level of a country's overall development in the domains of economy, education, and health)⁴³, number of hospital beds per 1000, month of data collection, government stringency level, and the number of new daily COVID-19 cases and deaths by the time of data collection as covariates to the analyses.

2. Similarly, I would suggest to the authors to include a control variable for the severeness of COVID-19 in the respective countries? For example, the number of cases/death per inhabitants.

Response: As noted above, we added rates for COVID-19 cases and deaths as covariates into our regression model. For more details, please refer to the copied excerpt (in response to the above comment).

3. Further, I would suggest to the authors to use a different regression specification for robustness purposes. Specifically, I would suggest using a specification including country fixed effects and clustering the standard errors across countries. This would help to address endogeneity concerns.

Response: We thank the reviewer for this suggestion which made us search for analytical methodologies to address endogeneity concerns in mixed-level regressions. However, we could not find out how to incorporate this suggestion into our analyses that we conducted with jamovi (which also produces a command that can be used in R). Some of the coauthors suggested to add 33 dummy variables for the 34 countries into the model; others suggested to cluster the countries into meaningful groups based on the study variables (e.g., via k-means), while a third group suggested to conduct analyses with only fixed effects. In the current version, we tested our predictions with a regression model where we specified fixed effects for the predictors' main effects and the two-way interactions, plus random effects for the predictors' main effects (country, trust, empathy and fear). Hence, due to my unfamiliarity with such kind of analyses (as it is rarely used in our discipline), we had trouble in clearly understanding what to do. If the Reviewer deems this analysis important, we would be happy to run these additional analyses and present them in the OSF files. However, for doing so, we would appreciate if the Reviewer can provide us with some clarification, assistance and guidance in how to do this (e.g., is there a command that can be added to the R for running these analyses?).

4. I like that the authors use the country-level scores for trust in the government from the World Values Survey. However, Engelhardt et al. (2021) also investigated a related question in the finance literature and used scores from the OECD for robustness checks. I would suggest to the authors to check robustness by using these scores as well.

Response: We thank the reviewer for this excellent suggestion which we have followed faithfully. When running the country-level analyses with the OECD trust scores (instead of WVS scores), we find a significant interaction in the hypothesized direction with empathy, but not with fear. We copied the analysis output towards the end of this letter. We believe that there may be several reasons for this. First, it should be noted that the sample decreases from 29 countries to 13 countries ($N = 5541$) when the OECD data is used, as much of the countries represented in our paper are not represented in the

OECD. As such, the OECD countries may represent a more selective and less comprehensive sample than the WVS countries which are more inclusive. Second, the OECD data on trust in the government more strongly reflects trust in times of COVID-19 while the WVS trust has been obtained from pre-pandemic times and would thus be more reflective of trust in government in a way that is more independent of the pandemic. Moreover, while the WVS trust and OECD trust scores were moderately correlated ($r = .29$), it is likely that they are substantially different. Correlational analyses showed that the OECD governmental trust score was more strongly associated with interpersonal trust from the WVS; also it was positively associated with a country's HDI while the trust score from the WVS was negatively associated with it. All this suggests that both types of assessment may measure different conceptualizations of trust in the government and thus not show parallel effects. While we believe that this additional robustness check may be valuable, we also thought that adding this analysis along with the discussion of the inconsistent results would spread the scope of the present paper. Hence, for now we decided not to include the OECD robustness check into the paper. However, if the Reviewer and the Editor consider this important, we may add this additional analysis along with its discussion into the paper (as we did with generalized trust).

5. I also recommend to the authors to check if there is a moderating effect when focusing the level of societal trust. In this respect, Engelhardt et al. (2021) note that “trust in governments might only be one side of the coin as trust in fellow citizens obeying the government's orders might also be of significant importance.” The data on societal trust is also available from the World Values Survey.

Response: We thank the Reviewer for this excellent suggestion. As already outlined above (in our response to the third comment 3 by Reviewer 2), we conducted additional analyses with generalized trust (both at the individual and country level), and reasoned about the obtained results in the discussion of the revised manuscript on page 25/26.

Page 25/26:

Another question to be addressed concerns the role of generalized trust (i.e., how much people trust each other in general), as much of our argumentation in regard to governmental trust seem also applicable to generalized trust, and since generalized trust has also been found to be a relevant predictor of compliance during the COVID-19 pandemic⁵². Yet —while we acknowledge the crucial role of generalized trust— the present research deemed important to focus on trust in the government, since pandemic management obviously takes more than just citizens' goodwill, and largely depends on the governments' activities and on how much people find them trustworthy⁵³. This is also in line with previous research, confirming the central and unique role that “systemic trust” has played for cooperative action during the COVID-19 pandemic, by showing that not only interpersonal trust ($r = 0.14, p < 0.05$) but also – and more strongly – trust in politicians was associated with COVID-19 vaccination ($r = 0.24, p < 0.01$)^{47,54,55}. Nevertheless, we conducted additional analyses to examine (1) whether our findings

remain robust when generalized trust is added as another covariate to the analyses, and (2) whether the proposed moderation effects can also be obtained with generalized (instead of governmental) trust, using the same analytical procedures as described above. The scores for individual-level generalized trust were obtained by averaging the scores that respondents provided to the following two questions: "In general, most people in our community can be trusted" and "Most people in our community are fair and do not take advantage of you" (1 = totally disagree, 5 = totally agree, $r = 0.78$). The scores for country-level generalized trust, were extracted from the most recent wave of the World Values Survey (WVS).

First, our analyses revealed that the results did not change when generalized trust was added as a covariate into the regression models. Second, when analyses were conducted with generalized instead of governmental trust, no interaction with fear and empathy emerged, neither on the individual-level nor the country-level analyses (except for a marginal significant interaction with fear of COVID-19 in the proposed direction with $p = 0.089$). Interestingly, generalized trust was also not associated directly with support for COVID-19 containment measures, neither on an individual level ($p = 0.073$) nor on a country level ($p = 0.204$). This supports the view that trust in the government seems conceptually distinct from generalized trust (in our research they were correlated with $r = 0.58$ on the individual-level; and $r = 0.10$ on at the country-level), exhibiting unique associations with support for COVID-19 containment behaviors (detailed results are available at <https://osf.io/kws9x/files/> under supplementary analyses).

Analysis Output OECD Data:

Results

Mixed Model

Model Info

Info	
Estimate	Linear mixed model fit by ML
Call	C19ContainmentBeh ~ 1 + EmpathicConcern + FearC19 + A1_Age + A2_Gender + HDI_TOTAL + Hosp_Beds + DataCol_Month + Gov_Stringency + New_Cases + New_Deaths + OECD_GovTrust + EmpathicConcern:FearC19 + OECD_GovTrust:EmpathicConcern + OECD_GovTrust:FearC19+(1 + EmpathicConcern + FearC19 + OECD_GovTrust Country)
AIC	10396.806
BIC	10582.164
LogLikel.	-5170.403
R-squared Marginal	0.311
R-squared Conditional	0.546
Converged	Yes
Optimizer	Bobyqa

Note. (Almost) singular fit. Maybe random coefficients variances are too small or correlations among them too large.

Note. boundary (singular) fit: see ?isSingular

Model Results

Fixed Effect Omnibus tests

	F	Num df	Den df	p
EmpathicConcern	46.1066	1	7.07	< .001
FearC19	81.5214	1	12.10	< .001
A1_Age	0.0589	1	5155.28	0.808
A2_Gender	1.2596	3	5514.25	0.286
HDI_TOTAL	32.9286	1	11.84	< .001
Hosp_Beds	7.7811	1	60.54	0.007
DataCol_Month	0.0409	1	10.50	0.844
Gov_Stringency	0.3302	1	11.53	0.577
New_Cases	0.8646	1	11.90	0.371
New_Deaths	0.7413	1	11.67	0.407
OECD_GovTrust	0.0127	1	11.68	0.912

Fixed Effect Omnibus tests

	F	Num df	Den df	p
EmpathicConcern * FearC19	12.0965	1	5502.62	< .001
EmpathicConcern * OECD_GovTrust	16.4050	1	8.49	0.003
FearC19 * OECD_GovTrust	0.2419	1	12.02	0.632

Note. Satterthwaite method for degrees of freedom

Fixed Effects Parameter Estimates

Names	Effect	Estimate	SE	95% Confidence Interval		Df	t	p
				Lower	Upper			
(Intercept)	(Intercept)	3.99339	0.07025	3.85570	4.13107	177.22	56.847	< .001
EmpathicConcern	EmpathicConcern	0.12876	0.01896	0.09159	0.16592	7.07	6.790	< .001
FearC19	FearC19	0.31788	0.03521	0.24887	0.38688	12.10	9.029	< .001
A1_Age	A1_Age	2.29e-4	9.44e-4	0.00208	0.00162	5155.28	-0.243	0.808
A2_Gender1	Female - Male	0.01776	0.01880	0.01909	0.05462	5507.42	0.945	0.345
A2_Gender2	none of the above - Male	0.02765	0.17847	0.32215	0.37745	5519.63	0.155	0.877
A2_Gender3	non binary - Male	0.26205	0.14689	0.02585	0.54995	5512.90	1.784	0.074
HDI_TOTAL	HDI_TOTAL	5.19093	0.90460	6.96392	3.41794	11.84	-5.738	< .001
Hosp_Beds	Hosp_Beds	0.01408	0.00505	0.00419	0.02397	60.54	2.789	0.007
DataCol_Month	DataCol_Month	0.00899	0.04447	0.09616	0.07817	10.50	-0.202	0.844
Gov_Stringency	Gov_Stringency	0.00261	0.00455	0.00630	0.01153	11.53	0.575	0.577
New_Cases	New_Cases	5.03e-4	5.41e-4	0.00156	5.57e-4	11.90	-0.930	0.371
New_Deaths	New_Deaths	0.02444	0.02839	0.03120	0.08009	11.67	0.861	0.407
OECD_GovTrust	OECD_GovTrust	0.00141	0.01246	0.02583	0.02302	11.68	-0.113	0.912
EmpathicConcern * FearC19	EmpathicConcern * FearC19	0.02575	0.00740	0.04026	0.01124	5502.62	-3.478	< .001
EmpathicConcern * OECD_GovTrust	EmpathicConcern * OECD_GovTrust	0.00429	0.00106	0.00222	0.00637	8.49	4.050	0.003

Fixed Effects Parameter Estimates

Names	Effect	Estimate	SE	95% Confidence Interval		Df	t	p
				Lower	Upper			
FearC19 *	FearC19 *	-	0.00360	-	0.00528	12.02	-0.492	0.632
OECD_GovTrust	OECD_GovTrust	0.00177	0.00360	0.00882	0.00528	12.02	-0.492	0.632

Random Components

Groups	Name	SD	Variance	ICC
Country	(Intercept)	0.0000	0.00000	0.00
	EmpathicConcern	0.0616	0.00379	
	FearC19	0.1191	0.01419	
	OECD_GovTrust	0.0408	0.00167	
Residual		0.6103	0.37247	

Note. Number of Obs: 5541 , groups: Country 13

Random Parameters correlations

Groups	Param.1	Param.2	Corr.
Country	(Intercept)	EmpathicConcern	NaN
	(Intercept)	FearC19	NaN
	(Intercept)	OECD_GovTrust	NaN
	EmpathicConcern	FearC19	0.4063
	EmpathicConcern	OECD_GovTrust	0.9375
	FearC19	OECD_GovTrust	0.0629

Simple Effects

Simple effects of EmpathicConcern : Omnibus Tests

Moderator levels				
OECD_GovTrust	F	Num df	Den df	p
Mean-1-SD	14.2	1.00	12.32	0.003
Mean	46.1	1.00	7.07	< .001
Mean+1-SD	66.4	1.00	11.50	< .001

Simple effects of EmpathicConcern : Omnibus Tests

Moderator levels

OECD_GovTrust	F	Num df	Den df	p
---------------	---	--------	--------	---

Simple effects of EmpathicConcern : Parameter estimates

OECD_GovTrust	Estimate	SE	95% Confidence Interval		Df	T	p
			Lower	Upper			
Mean-1-SD	0.0847	0.0225	0.0358	0.134	12.32	3.76	0.003
Mean	0.1288	0.0190	0.0840	0.174	7.07	6.79	< .001
Mean+1-SD	0.1728	0.0212	0.1264	0.219	11.50	8.15	< .001

Note. Simple effects are estimated keeping constant other independent variable(s) in the model

Effects Plots

27th Jul 23

Dear Dr Karakulak,

We apologise for the delay in processing your manuscript and thank you for your patience during the peer-review process. Your revised manuscript titled "Empathy, Fear of Disease and Support for COVID-19 Containment Behaviors: Evidence from 34 Countries on the Moderating Role of Governmental Trust" has now been seen by 2 reviewers, and I include their comments at the end of this message. The third reviewer (Reviewer #3) was unable to re-review your manuscript, so we asked Reviewer #2 to assess your responses to points 3 and 5 of Reviewer #3's report. Based on this assessment, you will see there are some outstanding points we would like you to address before making a final decision.

In particular, we ask you to conduct an analysis to check the robustness of your estimates to clustering (see second to last paragraph of Reviewer #3's additional comments).

We therefore invite you to revise and resubmit your manuscript, along with a point-by-point response to the reviewers. Please highlight all changes in the manuscript text file.

Editorially, we also ask you to use appropriate language to describe the null results. Statements such as 'Findings from the regression analysis suggest that country-level scores of trust in government did not interact with empathic prosocial concern ($p = 0.422$), which disconfirms Hypothesis 1b.' must be revised to read 'We found [no/little] credible evidence of X'.

Please use the following link to submit your revised manuscript, point-by-point response to the referees' comments (which should be in a separate document to any cover letter) and the completed checklist:

[Link redacted]

Please do not hesitate to contact me if you have any questions or would like to discuss these revisions further. We look forward to seeing the revised manuscript and thank you for the opportunity to review your work.

Best regards,

Antonia Eisenkoeck

Antonia Eisenkoeck
Senior Editor
Communications Psychology

EDITORIAL POLICIES AND FORMATTING

Editorial Policy: [Policy requirements](https://www.nature.com/documents/nr-editorial-policy-checklist.pdf) (Download the link to your computer as a PDF.)

Furthermore, please align your manuscript with our format requirements, which are summarized on the following checklist:

[Communications Psychology formatting checklist](https://www.nature.com/documents/commspsychol-style-formatting-checklist-article-rr.pdf)

and also in our style and formatting guide [Communications Psychology formatting guide](https://www.nature.com/documents/commspsychol-style-formatting-guide-accept.pdf) .

* **CODE AVAILABILITY:** All Communications Psychology manuscripts must include a section titled "Code Availability" at the end of the methods section. In the event of publication, we require that the custom analysis code supporting your conclusions is made available in a publicly accessible repository; at publication, we ask you to choose a repository that provides a DOI for the code; the link to the repository and the DOI will need to be included in the Code Availability statement. Publication as Supplementary Information will not suffice. We ask you to prepare code at this stage, to avoid delays later on in the process.

* **DATA AVAILABILITY:**

All Communications Psychology manuscripts must include a section titled "Data Availability" at the end of the Methods section or main text (if no Methods). More information on this policy, is available at <http://www.nature.com/authors/policies/data/data-availability-statements-data-citations.pdf>.

At a minimum the Data availability statement must explain how the data can be obtained and whether there are any restrictions on data sharing. Communications Psychology strongly endorses open

sharing of data. If you do make your data openly available, please include in the statement:

We recommend submitting the data to discipline-specific, community-recognized repositories, where possible and a list of recommended repositories is provided at <http://www.nature.com/sdata/policies/repositories>.

If a community resource is unavailable, data can be submitted to generalist repositories such as [figshare](https://figshare.com/) or [Dryad Digital Repository](http://datadryad.org/). Please provide a unique identifier for the data (for example a DOI or a permanent URL) in the data availability statement, if possible. If the repository does not provide identifiers, we encourage authors to supply the search terms that will return the data. For data that have been obtained from publicly available sources, please provide a URL and the specific data product name in the data availability statement. Data with a DOI should be further cited in the methods reference section.

REVIEWERS' COMMENTS:

Reviewer #1 (Remarks to the Author):

The authors have successfully addressed all my concerns. I congratulate them on a very nice paper.

Reviewer #2 (Remarks to the Author):

I thank the reviewers for extending their analyses to include generalised trust as a potential alternative source of variation. I agree with their conclusion that the effects they report in their manuscript are unique to institutional trust. As a side note, I find their observation that empathic concern is not correlated with generalised trust a rather interesting and puzzling one.

Additional comments by Reviewer #2:

As Reviewer #3 was unable to assess the revised version of your manuscript, we asked Reviewer #2 to assess points 3 and 5 from Reviewer #3's report.

They gave the following response:

I have now gone back to the manuscript and the reviewer reports.

Concerning R3 point 5: I do not seem to have access to the supplementary files where the analyses including social trust measures as a covariate and replicating the main analyses using social rather

than institutional trust are.

Because of my own interest in the matter, I however recall looking at those regressions and finding nothing to object.

Concerning R3 point 3: the authors seem to be confused as to how to implement fixed effects and clustered SE analyses as variants of their current models.

The fixed effects issue is simple: the authors should run simple OLS analyses including 33 dummies for 33 of the 34 countries for which they perform the analyses. This specification is somewhat similar in spirit to a multilevel random intercepts (not random slopes) model. The drawback of running this analysis is that it does not allow for random slopes (i.e. allowing each country to have its own coefficient of interest). Another important drawback of fixed effects models is that they do not allow the inclusion of country level regressors (such as for instance country specific covid19 mortality), which the authors rightly want to include.

While I agree with R3 that additional analyses allow to judge the robustness it seems to me that a fixed effects analysis in this case would lead to more information loss than gained insights.

For the clustering issue I can only give my intuition relying on my (perhaps wrong) understanding that the statistical packages the authors are using to do not allow for clustering. Let it also be clear that this is not my primary area of research, so my words should be taken with caution.

Background: clustered SE correct the estimated SE for correlation across sampling units (in this case, the countries). I would have called for clustering if the authors were running OLS regressions, but they are using multilevel regressions which by their nature allow for more complex data structures and are (to my understanding) more efficient than clustered SE OLS analyses. Many in fact suggest running random slopes multilevel models when clustering in OLS fixed effects analyses leads to large differences in the estimates. My intuition is therefore that the authors should be fine with what they are doing.

This being said, the most recent releases of the software STATA has a "mixed" command which runs multilevel regressions with any combination of random intercepts and random slopes, and which allows for clustered SE. Should the authors have access to it, they could run the analyses this way to check the robustness of their estimates to clustering.

On a side note, the authors might be interested in a recent draft looking at the effects of the covid crisis, the consequent regulation on behalf of the authorities and its consequences on both institutional and social trust levels, as the paper is very related to theirs. The manuscript can be found here: https://papers.ssrn.com/sol3/papers.cfm?abstract_id=4470394

Editor Comments:

1. In particular, we ask you to conduct an analysis to check the robustness of your estimates to clustering (see second to last paragraph of Reviewer #3's additional comments).

We thank the Editor and Reviewer 3 for their comment regarding the robustness of our results to a different regression approach. In the current version of the manuscript, the additional analyses are referred to on page 27/28; and details regarding the results of these analyses and a discussion of the results is presented in the Supplementary Materials.

Page 27/28:

Finally, we tested the robustness of our results when using a different methodological approach, namely a fixed effects regression model with cluster-robust standard errors. For this purpose, two additional analyses were performed: first, a simple OLS regression; and second, an OLS regression with the clustered standard errors correction. A table comparing the results obtained from these approaches, and a brief discussion concerning the differences in the results can be found in the Supplementary Materials (SM).

SM:

SM-Table 1. Comparison of Results under Different Regression Models

	Model 1: Initial model (Multi-level Modelling; MLM)	Model 2: Basic OLS regression	Model 3: OLS regression with cluster- robust standard errors
	B(SE)	B(SE)	B (SE)
Intercept	3.75 (0.09) ***	3.71 (0.01) ***	3.71 (0.08)***
TG (Country-Level)	0.01 (0.004)	0.01 (0.0001) ***	0.01 (0.004)*
EC	0.13 (0.01) ***	0.14 (0.01) ***	0.14 (0.01)***
FoC	0.31 (0.03) ***	0.30 (0.01) ***	0.30 (0.03)***
TG (Country-Lev.) × EC	-0.0003 (0.0003)	-0.0001 (0.0001)	0.0001 (0.0001)
TG (Country-Lev.) × FoC	-0.002 (0.001) **	-0.001 (0.0001) ***	-0.001 (0.001)
EC × FoC	-0.04 (0.01) ***	-0.03 (0.01) ***	-0.03 (0.01)**

Notes. TG = Trust in Government, EC = Empathic Concern, FoC = Fear of COVID-19. ** $p < 0.01$, *** $p < 0.001$. All analyses were performed by entering covariate effects of gender, age, HDI, hospital beds per 1000, month of data collection, government stringency level, and the number of new daily COVID-19 cases and deaths by the time of data collection. For reasons of simplicity, the covariate effects are not displayed in the Table.

The comparison of the results obtained under the three different methodological approaches revealed that results from the OLS regression approach (Model 1) confirmed the results that we obtained when using a random-slopes MLM approach (Model 1). In both types of regression models, the interaction between trust in the government on the country-level and fear of COVID-19 was found significant, while the statistical analyses did not provide evidence for a significant interaction between country-level trust in the government and empathic prosocial concern. However, when applying the cluster-robust standard errors

correction to the OLS regression (Model 3), the results obtained from this analysis did no longer support the existence of a significant interaction between trust in the government and fear of COVID-19. Hence, while Hypothesis 2b seems supported under Model 1 and Model 2, the analysis results obtained under Model 3, where the size of standard errors is typically larger, do not support this hypothesis.

While this difference in results between Model 1 and Model 3 may be interpreted as an indicator for a less robust interaction effect, it should also be noted that there is an ongoing discussion concerning the use of these two regression approaches (MLM vs. OLS with cluster-robust errors). Whether one or the other approach should be preferred pretty much depends on the research setting and the type of research question that is to be tested. While the OLS with cluster-robust standard errors seems more appropriate for single-level models where the researcher theorizes variation across different units (i.e., countries) more like a nuisance factor that must be controlled for, the MLM approach seems more appropriate in settings that assume a multi-level structure where the different units are more independent from each other and can produce different effects (for a comparison of these two approaches, see¹). We thus believe that the random-effects MLM approach (Model 1) that we reported in the main analyses is more suitable for testing the hypotheses in the current research setting.

References

1. Oshchepkov, A., & Shirokanova, A. Bridging the gap between multilevel modeling and economic methods. *Social Science Research* 104, 102689 (2022).
2. Editorially, we also ask you to use appropriate language to describe the null results. Statements such as ‘Findings from the regression analysis suggest that country-level scores of trust in government did not interact with empathic prosocial concern ($p = 0.422$), which disconfirms Hypothesis 1b.’ must be revised to read ‘We found [no/little] credible evidence of X’.

We thank the Editor for raising attention to this issue. We carefully screened the manuscript for any null-result and made sure to update the formulation of these results in such a way that it fits with the journal’s guidelines. All updates (both in the main manuscript and the supplementary materials are presented below).

P. 27:

Again, supporting COVID-19 containment behaviors was significantly associated with both empathic prosocial concern and fear of COVID-19 (both $ps < 0.001$, Cohen’s $f = 3.41$ for empathic concern and 2.00 for fear of COVID-19), while no evidence was found for a significant association with country-level trust in the government ($p = 0.69$).

The findings from the regression analysis did not support a significant interaction between country-level scores of trust in government and empathic prosocial concern ($p = 0.422$), which disconfirms Hypothesis 1b.

SM:

Again, the analysis results did not change, and confirmed that trust in the government moderated the association between fear of COVID-19 and support for COVID-19

containment behaviors ($\beta = -0.002$, $t(26.8) = -2.09$, 95% CI [-0.004, -0.0001], $p = 0.047$, Cohen's $f = 0.34$); while we found no evidence for a significant moderation for the association between empathic concern and support for COVID-19 containment behaviors ($p = 0.341$).

The results for this regression model did not provide evidence for a significant interaction; neither between generalized trust and empathic concern ($p = 0.95$), nor between generalized trust and fear of COVID-19 ($p = 0.95$). Second, the same analyses were repeated on the country-level with generalized trust scores extracted from the World Values Survey (see step 4 in Table 5). Again, the results obtained from the regression analysis did not provide evidence for the proposed interactions (H1b and H2b). Both the interaction between generalized trust and empathic concern ($p = 0.22$) and between generalized trust and fear of COVID-19 ($p = 0.09$) were not found as significant.

Overall, we are grateful for the excellent guidance provided by the Editor and the Reviewers, and their helpful approach throughout the revision process. We hope that with its revisions, the current version of the manuscript meets your expectations and would be very happy if it will be accepted for publication. Of course, we are also happy to incorporate any additional suggestions that would further improve the quality of our paper.

Sincerely (on behalf of all co-authors),
Arzu Karakulak

6th Oct 23

Dear Dr Karakulak,

Your manuscript titled "Empathy, Fear of Disease and Support for COVID-19 Containment Behaviors: Evidence from 34 Countries on the Moderating Role of Governmental Trust" has now been seen by our reviewers, whose comments appear below. In light of their advice I am delighted to say that we are happy, in principle, to publish a suitably revised version in Communications Psychology under the open access CC BY license (Creative Commons Attribution v4.0 International License).

We therefore invite you to revise your paper one last time to address the remaining concerns of our reviewers and a list of editorial requests. At the same time we ask that you edit your manuscript to comply with our format requirements and to maximise the accessibility and therefore the impact of your work.

Please note that it may still be possible for your paper to be published before the end of 2023, but in order to do this we will need you to address these points as quickly as possible so that we can move forward with your paper.

EDITORIAL REQUESTS:

Please address all formatting requests precisely, including additional checks that all links work (and are not affected by typos) and that each statement contains the right information and is placed in the right location in the manuscript file. We ask you to adhere to the exact wording that's required for positive and negative statements, for example in the context of your conflict of interest statement, which should be global (if negative), and specific (if positive). At the same time, we ask you to avoid unnecessary duplication of information across the manuscript.

SUBMISSION INFORMATION:

OPEN ACCESS:

Communications Psychology is a fully open access journal. Articles are made freely accessible on publication under a [CC BY](http://creativecommons.org/licenses/by/4.0) license (Creative Commons Attribution 4.0 International License). This license allows maximum dissemination and re-use of open access materials and is preferred by many research funding bodies.

For further information about article processing charges, open access funding, and advice and support from Nature Research, please visit <https://www.nature.com/commspsychol/article-processing-charges>

At acceptance, you will be provided with instructions for completing this CC BY license on behalf of all authors. This grants us the necessary permissions to publish your paper. Additionally, you will be asked to declare that all required third party permissions have been obtained, and to provide billing information in order to pay the article-processing charge (APC).

* **DATA AVAILABILITY:**

[Link redacted]

Best regards,

Antonia Eisenkoeck

Antonia Eisenkoeck
Senior Editor
Communications Psychology

REVIEWERS' EXPERTISE:

Reviewer #1

Reviewer #2

REVIEWERS' COMMENTS:

Reviewer #2 (Remarks to the Author):

Thank you for your work on the article and for its contribution.

All my comments have been addressed carefully, and I have no further remarks on the draft.